# A window of opportunity for cooperativity in the T Cell Receptor

N. Martin-Blanco[1], R. Blanco[1], C. Alda-Catalinas [1], E. R. Bovolenta[1], C.L. Oeste[1], E. Palmer[2], W.W. Schamel[3,4,5], G. Lythe[6], C. Molina-París[6], M. Castro[6,7] & B. Alarcon[1]

The T-cell antigen receptor (TCR) is pre-organised in oligomers, known as nanoclusters. Nanoclusters could provide a framework for inter-TCR cooperativity upon peptide antigen-major histocompatibility complex (pMHC) binding. Here we have used soluble pMHC oligomers in search for cooperativity effects along the plasma membrane plane. We find that initial binding events favour subsequent pMHC binding to additional TCRs, during a narrow temporal window. This behaviour can be explained by a 3-state model of TCR transition from Resting to Active, to a final Inhibited state. By disrupting nanoclusters and hampering the Active conformation, we show that TCR cooperativity is consistent with TCR nanoclusters adopting the Active state in a coordinated manner. Preferential binding of pMHC to the Active TCR at the immunological synapse suggests that there is a transient time frame for signal amplification in the TCR, allowing the T cells to keep track of antigen quantity and binding time.

[1] Centro de Biología Molecular Severo Ochoa, Consejo Superior de Investigaciones Científicas, Universidad Autónoma de Madrid, 28049 Madrid, Spain. [2] University Hospital Basel, Hebelstrasse 20, 4031 Basel, Switzerland. [3] Faculty of Biology, Institute Biology III, University of Freiburg, 79104 Freiburg, Germany. [4] Centre for Biological Signalling Studies (BIOSS), University of Freiburg, 79104 Freiburg, Germany. [5] Center for Chronic Immunodeficiency (CCI), Medical Center Freiburg and Faculty of Medicine, University of Freiburg, 79104 Freiburg, Germany. [6] School of Mathematics, University of Leeds, Leeds LS2 9JT, UK. [7] Grupo Interdisciplinar de Sistemas Complejos (GISC), Universidad Pontificia Comillas, Alberto Aguilera25, 28015 Madrid, Spain. Correspondence and requests for materials should be addressed to C.M-Pís. (email: carmen@maths.leeds.ac.uk) or to M.C. (email: marioc@comillas.edu) or to B.A. (email: balarcon@cbm.csic.es)

T cells contain low-affinity receptors (T-cell receptors, TCRs) that nonetheless achieve high specificity and sensitivity for antigen peptide/MHC (pMHC) ligands[1]. This paradox is exacerbated when considering that the difference in affinity for pathogen-derived pMHC versus self-pMHC complexes is small enough to be compensated by the law of mass action. A hypothetical explanation is that TCRs are pre-organised in nanoclusters of up to 20 TCRs that could provide a framework for inter-TCR cooperativity upon pMHC binding[2–6].

The TCR is composed of six subunits (TCRα, TCRβ, CD3γ, CD3δ, CD3ε and CD3ζ) without intrinsic enzymatic activity, but functionally associated to cytoplasmic tyrosine kinases[7]. Using monovalent versus multivalent fragments of activating antibodies and monomeric and multimeric forms of recombinant soluble pMHC, it was found that simultaneous binding of two or more TCRs by the ligands is required for TCR triggering[8–11]. Since the TCR appears to be organised in nanoclusters before antigen binding[3–6], the need for bivalent or multivalent binding of the pMHC ligand must not rely on promoting dimerisation or multimerization, for TCR nanoclusters are already oligomeric. Instead, we found that ligand-mediated TCR crosslinking is required to stabilise the TCR in its Active conformation[11], opening the possibility of allosteric regulation within nanoclusters.

Allostery is intrinsic to the control of metabolic and signal-transduction pathways. It is defined in functional terms as a comparison of how a ligand binds in the presence or absence of an already bound first ligand[12]. Membrane receptors offer examples of allostery, as is the case of ligand binding to the ectodomain of seven transmembrane receptors[13]. Upon binding, transmission of information across the membrane to the cytoplasm favours the binding of a signalling G protein to a distal site. Furthermore, this information is also transmitted along the plane of the membrane, resulting in the formation of receptor homodimers or heterodimers that affect binding of a second extracellular ligand[14].

In this context, we have now approached the study of pMHC ligand binding to the TCR in search of homotropic allosteric effects along the plane of the plasma membrane. We provide evidence suggesting the existence of cooperativity upon ligand binding. However, we found that this cooperativity peaks 4–8 min after TCR engagement by the first pMHC ligand and decays thereafter. This delineates a time window in which signal amplification could operate by favouring additional pMHC ligand binding to TCRs in the same nanocluster. Considering the transition of the TCR between three activation states, we propose a model for regulation of T-cell activation in physiological conditions. By means of mathematical modelling, we show that the observed effects are consistent with cooperativity among receptors in nanoclusters and the coexistence of three allosteric configurations with different functional properties.

## Results

**A time optimum for Active TCR.** The monoclonal antibody APA1/1 detects a conformational epitope in the cytoplasmic tail of CD3ε (Fig. 1a)[15]. This epitope lies within the proline-rich sequence in CD3ε and becomes exposed when the TCR is triggered by binding to an activating ligand. We have studied how APA1/1 epitope is exposed upon stimulation of CD8[+] T cells from OT-1 TCR transgenic mice with a soluble H-2K[b] tetramer loaded with the strong TCR agonist OVAp (SIINFEKL; OVAp tetramer from now on). We first titrated the concentrations of OVAp tetramer that are optimal for the activation of OT-1 T cells 24 h after stimulation, measured by the induction of CD69 and CD25 expression. Expression of both activation markers peaked

at concentrations of 1–10 nM. Higher doses did not improve the response (Fig. 1b) or even worsened it (Supplementary Fig. 1a). Interestingly, OVAp tetramer concentrations leading to maximum T-cell activation (1–10 nM) was well below that needed for saturation of binding (above 1000 nM). Titration curves for the OVAp tetramer were also performed by incubation for a shorter time at 0 °C, confirming that concentrations higher than 1000 nM are needed to saturate all binding sites on T cells (Fig. 1c). When OVAp tetramer binding at a sub-saturating concentration (1 nM) at 0 °C was plotted as a function of time, we found that maximum binding was not reached even after 20 min of incubation (Fig. 1d). However, when the exposure of APA1/1 epitope was plotted, we found a maximum at 6 min, with a decline afterwards, even though binding of the OVAp tetramer continued to increase. Although binding of 1 nM OVAp tetramer had not reached a plateau after 20 min of incubation, the use of a 200-fold higher concentration of tetramer (200 nM) accelerated the binding and a plateau was reached after 8 min of incubation (Fig. 1d). Interestingly, the APA1/1 optimal binding time point was independent of the concentration of tetramer (1 versus 200 nM) used to induce the exposure of the APA1/1 epitope. The time maximum was also unaltered when incubations were carried out at 37 °C, indicating that the molecular processes leading to the skewed bell-shaped curve of APA1/1 epitope exposure were largely temperature-independent and therefore independent of the cell's metabolism (Fig. 1e). Control experiments were also carried out to assess TCR specificity towards OVAp tetramer binding and tetramer-triggered exposure of the APA1/1 epitope. OT-1 CD8 + T cells were stimulated with 1 nM OVAp tetramer, in parallel to stimulation of CD8 + T cells from another mouse line, which bears a TCR transgene of irrelevant specificity (HY TCR). Both tetramer binding and time-dependent APA1/1 exposure were specific of the OT-1 TCR (Fig. 1f and Supplementary Fig. 1b). In order to exclude that the binding properties of the OVAp tetramer were altered by the presence of tetramer aggregates, we analysed the OVAp-APC tetramer by size-exclusion chromatography. The MHC-I heavy chain peaked in fractions between 1.5 and 2 ml (Supplementary Fig. 1c) which presented maximum binding activity in OT-1 CD8 + T cells (Supplementary Fig. 1d). The molecular weight of these fractions was between 150 and 669 kDa, in the range of the expected size for the OVAp-APC tetramer (~242 kDa tetramer + ~105 kDa APC = ~347 kDa), not far from previous gel filtration results[16] and contaminating trimers, dimers and monomers[17]. No MHC-I heavy chain or binding activity was detected in fractions < 1 ml, indicating the absence of large aggregates in the OVAp tetramer preparation (Supplementary Fig. 1c and 1d).

The data above highlight that there is an optimum time point for APA1/1 epitope exposure, indicative of Active conformation of the TCR when CD8 + T cells are stimulated with MHC-I tetramer concentrations well below saturation.

**A 3-state model can explain the time optimum for Active TCR.** In order to get insight into the time-dependent exposure of APA1/1 epitope, we first assumed a 2-state model (Resting and Active, Fig. 1a and Fig. 2a), in which ligand binding stabilises the Active state and this effect lasts as long as the pMHC ligand is bound to the TCR. With this 2-state model, a plateau of TCR in the Active conformation is reached and does not decrease (Fig. 2b). However, a 3-state conformational model would successfully account for the decreasing behaviour of the curve after 6 min of incubation. Each of the three states corresponds to ensembles of different allosteric configurations. The Resting state is formed by the ensemble of all conformations, in which neither the extracellular domains of the TCRα/β subunits are in an

optimal orientation for binding nor are the cytoplasmic CD3 tails prone to binding effector proteins. The Active state is formed by the ensemble of conformations optimal for pMHC binding to TCRα/β in all TCRs, as well as for cytoplasmic effector binding by CD3 subunits in all TCRs. Lastly, the Inhibited state is formed by the ensemble of conformations in which some TCRα/β ectodomains in the nanoclusters favour pMHC binding and others do not, whereas the CD3 cytoplasmic tails are not exposed to intracellular effectors (Fig. 2a). For the sake of simplicity, we

develop a mathematical model of a nanocluster composed of three TCRs and a trimeric pMHC ligand, which is in fact the functional valence of a tetramer (Fig. 2c). A model that includes larger nanoclusters will only change our results in a quantitative, but not qualitative way[18].

We describe the free-energy landscape of the three conformational states in the absence and presence of ligand-induced crosslinking in Supplementary Fig. 2a and a summary of the transitions between states with and without ligand in Fig. 2c.

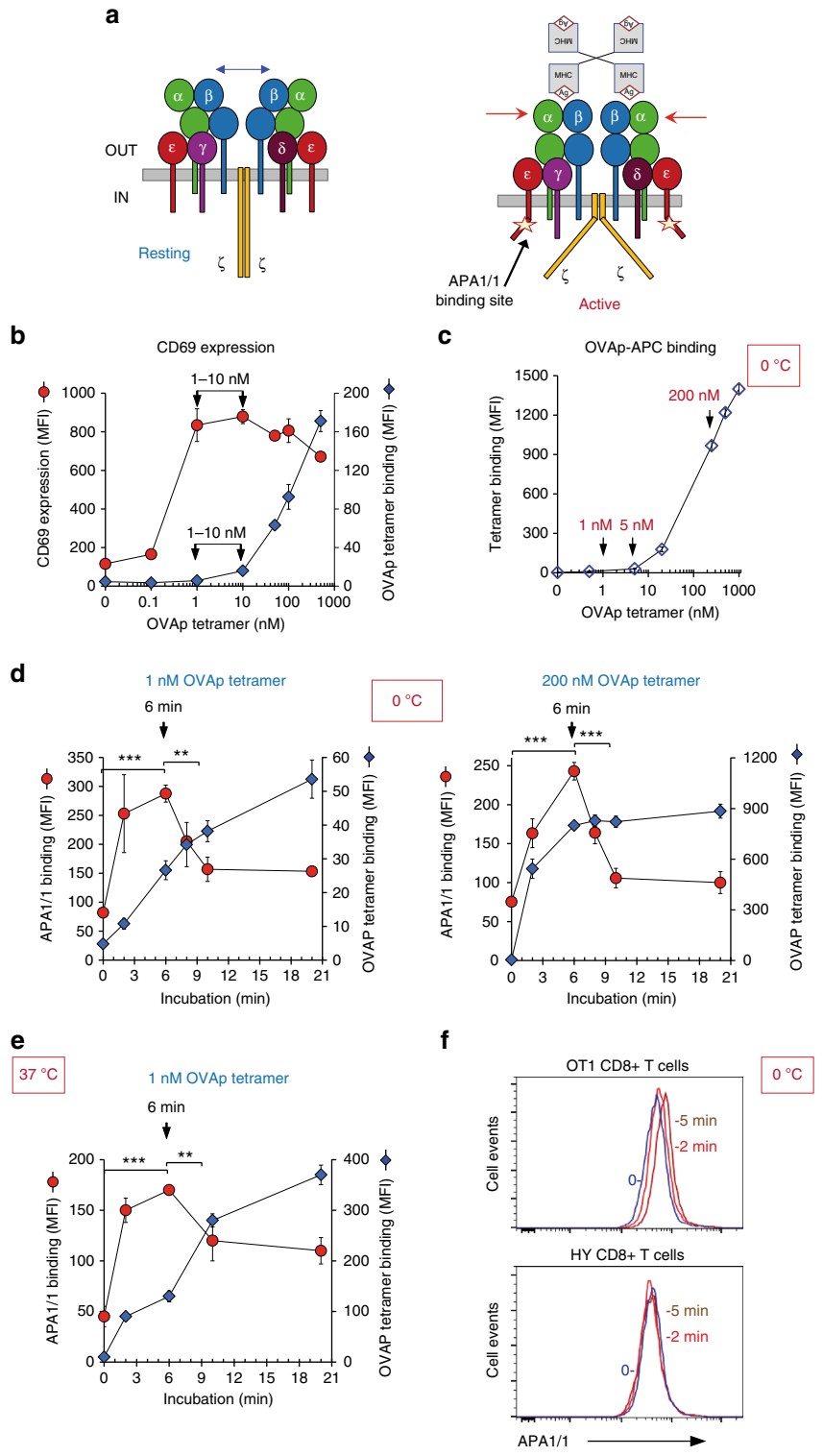

The mathematical model captures the free-energy changes induced by ligand crosslinking and the transition rates between each nanocluster state. It assumes that single pMHC-mediated crosslinking is sufficient to stabilise the Active state of the entire TCR nanocluster, favouring pMHC binding to all TCRs in the cluster (Fig. 2d). The model is consistent with the experimental data generated with APA1/1 (Fig. 1d), showing that the number of nanoclusters in the Active state displays a time-dependent maximum (Fig. 2e). The location of this maximum is almost insensitive to parameter changes, such as ligand concentration, temperature and free-energy differences between the three states (Supplementary Fig. 2b and Supplementary Note 1).

We can now enquire the model to decide if the existence of the maximum in the number of TCR nanoclusters in the Active state has further immunological consequences. Given the decay of the Active state (as well as the increase of the Inhibited state) shown in Fig. 2e, the model predicts that the enhancement in subsequent binding events is a transient effect. This effect can be explored if we add the first dose of pMHC ligand (preincubation), and at a fixed later time we add a second. In Fig. 2f, we show that binding of the second ligand is maximised when the preincubation time is around 5 min, which is the timescale needed for the Active state to reach its maximum (Fig. 2e).

**Ligand binding cooperativity suggests inter-TCR crosstalk.** Following the predictions of the model, we searched for signs of cooperativity as revealed by the properties of binding to the pMHC ligand. Thus, we measured the effect of preincubation of OT-1 T cells (at 0 °C) with a sub-saturating concentration of APC-labelled OVAp tetramer (1 nM) on subsequent binding of the same tetramer, but in a different colour (PE), for a fixed time of 5 min (Fig. 3a). As expected, binding of the OVAp-APC tetramer used for preincubation steadily increased with its time of incubation (Fig. 3a, blue diamonds). In sharp contrast, binding of the second tetramer (OVAp-PE) during the 5 min time window was enhanced if the cells had been preincubated with the first tetramer, with a maximum at 4–6 min of preincubation (Fig. 3a, red circles). To better adjust the maximum for positive cooperativity upon ligand binding, we carried out a similar experiment by reducing the incubation period with the second tetramer to 1 min. We found a sharp raise in the binding of the second tetramer upon 4 min of preincubation that slowly decayed from 4 to 6 min of preincubation and decayed faster from 6 min onwards (Supplementary Fig. 3a). These results document the existence of cooperativity phenomena within the TCRs manifested upon ligand binding, i.e. binding of the pMHC ligand to a few TCRs improves the subsequent binding of additional pMHC ligands to free TCRs, albeit during a narrow time window. The effect could be reproduced using CD8$^+$ T cells from double TCR transgenic mice that bear two TCRs (OT-1 and HY) of different antigen specificity. Preincubation with OVAp-APC tetramer, specific for OT-1, exerted a time-dependent cooperative effect on the binding of a second HYp-PE tetramer specific for the HY TCR (Fig. 3b). A similar effect was observed when the order of tetramer addition was inverted (Supplementary Fig. 3b). The above results show that clustered TCRs cooperate in the following sense: a conformational change in part of the cluster affects the whole nanocluster and, subsequently, an increase in the number of clusters in the Active state enhances ligand binding.

Since pMHC interacts with both the TCR and the coreceptors (CD8 in CD8 + T cells), we interrogated if the transient positive cooperativity effect upon ligand binding was mediated by binding to CD8. We found that blocking CD8 using an anti-CD8 antibody strongly inhibited the intensity of OVAp tetramer binding, but did not abolish the time-dependent cooperativity effect (Supplementary Fig. 4a). Likewise, an OVAp tetramer bearing a point mutation that prevents binding to CD8, but not to the TCR, was also able to induce the transient cooperativity effect on binding of a second wild-type OVAp tetramer (Supplementary Fig. 4b).

**Ligand binding cooperativity requires TCR crosslinking.** The pMHC tetramers we used are tetrahedral, with a pMHC–pMHC distance in the tetramer of ~80 Å; higher than the optimal distance required for effective binding to the T-cell surface[19, 20]. To determine if ligand binding cooperativity was related to the geometry of the pMHC oligomer, we studied binding of fluorescent pMHC oligomers based on a linear dextran frame. In analogy to the pMHC tetramers, we found that preincubation with an OVAp-PE dextramer had an impact on the binding of a second OVAp-APC dextramer in a time-dependent manner (Supplementary Fig. 5a). These results suggest that the exact arrangement of the pMHC ligands in the stimulating pMHC oligomer do not seem to be relevant. However, ligand-mediated TCR crosslinking was required because preincubation with an OVAp monomer did not influence subsequent binding of a tetramer either in a time-dependent fashion (Fig. 4a) or in a concentration-dependent manner (Fig. 4b). However, preincubation with the OVAp tetramer exerted a time-dependent cooperative effect on the binding of the OVAp monomer (Fig. 4c). Likewise, preincubation with the OVAp tetramer for 6 min favoured concentration-dependent binding of the monomer (Fig. 4d). These data suggest not only that TCR crosslinking is required to induce cooperative binding of subsequent pMHC ligands, but also that ligand binding cooperativity takes place by favouring the exposure of individual pMHC-binding sites on individual TCRs within nanoclusters in the Active conformation.

**Cooperativity requires nanoclusters to adopt the Active state.** Tetramer and dextramer binding experiments were carried out at 0 °C, which decreases the possibility that the first phase of

**Fig. 1** Optimum time for Active conformation of the TCR at sub-saturating concentrations of the pMHC ligand. **a** Cartoon of the Resting and ligand-bound Active conformations of the TCR. Bivalent or multivalent ligation of two or more TCRs by its pMHC ligand results in an outside-in transmission of a conformational change detected in the cytoplasmic tails of the CD3 subunits. One of these changes consists in exposing an epitope in the proline-rich region of CD3ε for mAb APA1/1 binding. **b** OT-1 CD8 + T cells were incubated at 37 °C with the indicated concentrations of APC-labelled OVAp tetramer for 24 h. After stimulation, cells were stained with anti-CD69 and analyzed by cytometry. The mean fluorescence intensity (MFI) for CD69 staining is shown as red circles and that of tetramer binding as blue diamonds. **c** OT-1 CD8 + T cells were incubated on ice with the indicated concentrations of APC-labelled OVAp tetramer for 40 min. MFI was calculated by flow cytometry. **d** OT-1 CD8 + T cells were incubated at different times at 0 °C with 1 nM or 200 nM of OVAp-APC tetramer and subsequently fixed, permeabilized and stained with the APA1/1 mAb prior to flow cytometry analysis. MFI for APA1/1 staining is shown as red circles and that of tetramer binding as blue diamonds. **e** OT-1 CD8 + T cells were incubated at different times at 37 °C with 1 nM OVAp-APC tetramer and stained with the APA1/1 mAb as in Fig. 1d. **f** Histogram overlay for APA1/1 expression in OT-1 and HY CD8 + T cells not incubated (blue lines) or incubated for 2 min (red line) or 5 min (brown line) with 1 nM OVAp-APC tetramer at 0 °C. All data shown in Fig. 1 represent the mean ± s.d. of triplicate datasets; *$p < 0.05$; **$p < 0.005$; ***$p < 0.0005$ (two-tail unpaired $t$-test)

cooperativity could be due to unexpected aggregation of TCR nanoclusters induced by preincubation with the first tetramer. This could have increased the valence of the TCR for subsequent binding events. However, this hypothesis is at odds with the negative slope of the kinetic curves observed typically after 4–8

min of preincubation. Notwithstanding, we examined the distribution of the TCR on the surface of OT-1 T cells after incubation with OVAp tetramer at 0 °C and 37 °C. Confocal microscopy revealed the formation of an aggregated cup structure in T cells incubated with the tetramer at 37 °C, but not when

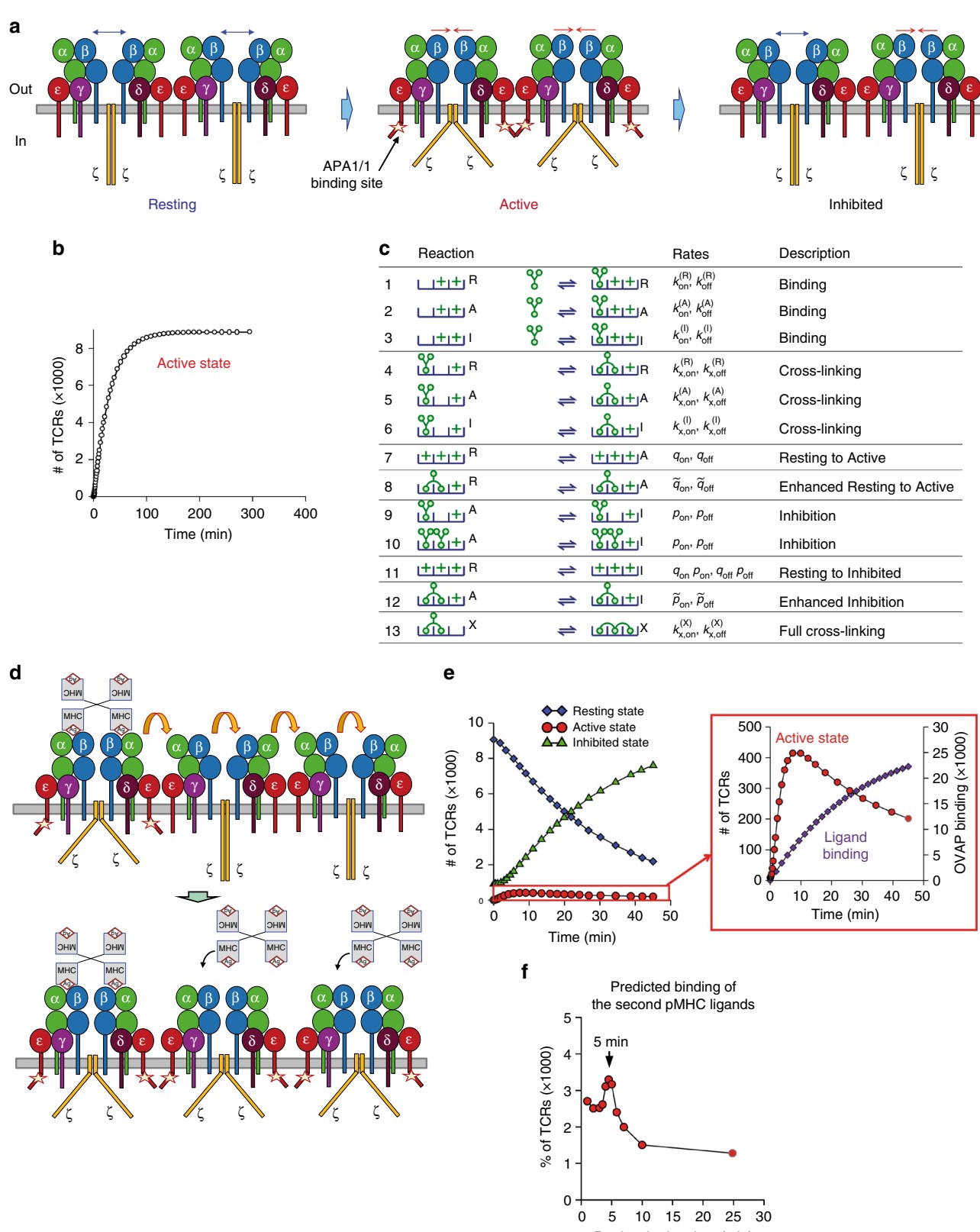

incubated at 0 °C (Supplementary Fig. 6a). We also used immunogold and label-fracture electron microscopy to determine if the size of the pre-existing TCR nanoclusters was altered upon incubation with OVAp tetramer. We found that while incubation at 37 °C induced the formation of large 2-dimensional TCR clusters of even more than 30 TCRs, incubation at 0 °C did not alter the size of the pre-existing nanoclusters (Supplementary Fig. 6b).

We then investigated if the observed cooperativity upon ligand binding depended on the existence of TCR nanoclusters. Since TCR nanoclusters require cholesterol[5, 21], we first analysed the effect of cholesterol extraction with methyl-β-cyclodextrin (MβCD) on ligand binding cooperativity. We found that treatment with MβCD did not alter TCR abundance in the plasma membrane evaluated by incubation with an anti-TCRβ antibody (Fig. 5a, left). However, cholesterol depletion completely abrogated the cooperativity effects of OVAp-APC preincubation on OVAp-PE binding (Fig. 5a, right) and also dramatically reduced time-dependent binding of OVAp-APC (Fig. 5a, centre), suggesting that TCR nanoclusters were required for cooperativity.

We next tested if the transient positive effect upon ligand binding was associated with the adoption of the Active conformation. We have previously described a point mutation (C80G) in the stalk sequence of the extracellular domain of CD3ε (Fig. 5b) that prevents the outside-in transmission of conformational changes, which would normally result in the exposure of the proline-rich sequence in the cytoplasmic tail of CD3ε for Nck binding[22]. This mutation abrogates αβ T-cell development in the thymus. Nevertheless, the expression of a TCR transgene partly rescues this defect, and nearly normal numbers of mature CD8[+] T cells bearing the OT-1 TCR are found in lymphoid organs of mice homozygous for the CD3ε C80G mutation[22]. However, since these cells express lower amounts of TCR than their wild-type counterparts, we tested the effect of the C80G mutation on OVAp tetramer ligand binding by gating T-cell populations with equal TCR expression in WT or C80G mutant OT-1 cells (Fig. 5c, left). Binding of increasing amounts of OVAp tetramer to the WT population was normalised to match TCR expression in C80G T cells (WT_{norm}) and compared with binding to the total WT population. It was then evident that the difference in TCR expression (threefold higher in the WT) did not significantly affect the tetramer binding. However, the C80G mutation inhibited binding, compared with both the unselected and selected WT populations (Fig. 5c, right). In addition, the C80G mutation strongly inhibited the transient enhancement in OVAp-PE tetramer binding promoted by OVAp-APC tetramer preincubation (Fig. 5d). These results show that the TCR needs to adopt the Active conformation in order for TCR nanocluster cooperativity to take place. Importantly, the C80G mutation is in the ectodomain of CD3ε, whereas the pMHC ligand binding site is formed by the V regions of TCRα and TCRβ. Therefore, abrogation of binding cooperativity by the C80G mutation suggests that a transmission of structural rearrangements from engaged TCRα/β to non-engaged TCRα/β is mediated by CD3ε and perhaps other CD3 subunits (Fig. 2d).

**Ligand binding changes the orientation of TCRα/β ectodomains.** We then searched for evidence that could reflect structural rearrangements within TCR nanoclusters that could be responsible for the observed cooperativity upon ligand binding. For this, we studied fluorescence resonance energy transfer (FRET) between PE-labelled and APC-labelled tetramers upon simultaneous binding to OT-1 T cells as a function of time. FRET intensity between both tetramers was compared on one hand with that of a positive control, i.e. energy transfer between the TCR-bound tetramer and a labelled anti-CD3ε antibody. On the other hand, the negative control was set as the energy transfer between the TCR-bound tetramer and anti-CD27 antibody, which is a membrane protein not described as a TCR interactor (Fig. 6a, Supplementary Fig. 7a). Binding of both tetramers followed a similar time-course (Supplementary Fig. 7b). A set of controls of the FRET assay is shown in Supplementary Fig. 7c. At the end of the incubation period (20 min) at 0 °C, FRET intensity between the two bound tetramers was higher than that of the negative control and even than that of the positive control (Fig. 6b, left panel). Higher FRET intensity registered for two bound tetramers than for the positive control (a tetramer and anti-CD3) is likely reflecting a more favourable orientation for FRET. This is probably due to the fact that two tetramers bind to the tips of two TCRα/β ectodomains, whereas the anti-CD3 binds the ectodomain of CD3ε in a diagonal outward orientation[23, 24]. The fact that there is a considerable FRET signal between two binding tetramers itself reflects the existence of preformed TCR nanoclusters.

FRET intensity increased with time following tetramer binding, much like FRET intensities for the positive and negative controls. However, both FRET intensity and FRET efficiency between tetramers experienced a sudden increase between 5 and 8 min that was not detected in the controls (Fig. 6b, arrows and Supplementary Fig. 7b). These data indicate that a qualitative shift in the proximity or orientation of different TCRs is produced 5 to 8 min after the first pMHC tetramer ligands start binding.

The shift in tetramer–tetramer FRET intensity suggested a change in the distance, or in the orientation, of the TCRα/β heterodimers, since they contain the binding site for pMHC. To address this hypothesis, we assessed if low sub-saturating concentrations of OVAp tetramer modified TCRα–TCRβ FRET measurements. We used unlabelled OVAp tetramer and fluorescent anti-Vα2 and anti-Vβ5 antibodies which recognise the

**Fig. 2** TCRs are in transit between three different states of conformations. **a** Cartoon of the Resting, Active and Inhibited states. Blue and red arrows suggest a possible movement of the αβ ectodomains. Adoption of the three states is also reflected by movements of the cytoplasmic tails of the CD3 subunits. One of these movements results in exposure of the APA1/1 epitope in CD3ε. **b** A mathematical model with just two states (Resting and Active) cannot explain the optimum time point for the Active conformation shown in Fig. 1d, e. Rather, Active conformation reaches a plateau of indefinite duration. **c** Summary of reactions included in the model. The letters R, A and I stand for the states Resting, Active and Inhibited. A nanocluster of three TCRs and a trimeric pMHC ligand were chosen for modelling. The numerical values of the parameters and an explanatory description of the states are summarised in Methods. The plus symbols stand for either empty or non-crosslinked bound ligand used for conciseness. **d** Model to explain how ligation of two or more TCRs by pMHC results in stabilisation of the Active conformation in the entire TCR nanocluster. Adoption of the Active conformation by unbound TCRs facilitates their binding to additional pMHC ligands. **e** Numerical integration of the 3-state model after addition of ligand. The right panel shows the existence of an optimum of the Active conformation (red circles), even though binding has not yet reached saturation. This behaviour reflects the experimental results in Fig. 1d–e. **f** Prediction of the existence of cooperative effects on pMHC ligand binding derived from the mathematical model. Preincubation of TCR nanoclusters with free ligand at different times displays an optimum cooperation at the time when the maximal number of TCR nanoclusters in the Active conformation is found (Figs. 1d, e and 2e)

variable regions of the OT-1 TCR. Since the tetramer and the anti-V region antibodies are likely to compete for binding to the TCR, the effect of tetramer binding would be measured on nearby unoccupied TCRα/β heterodimers in the nanocluster, in the sub-saturating conditions of tetramer employed (Fig. 6c). The anti-Vα2 probe can act as a FRET donor for anti-Vβ5 probe bound to the same TCR or bound to a different nearby TCR. For this reason, we carried out FRET analysis in cells pretreated or not with MβCD to disrupt nanoclusters. We detected an increase of FRET efficiency with an optimum of 4 min that was not apparent in T cells pretreated with MβCD (Fig. 6d). Interestingly, FRET efficiency in cells not preincubated with tetramer (0 min, Fig. 6d) was not affected by MβCD, suggesting that the FRET signal that is resistant to the disruption of nanoclusters is generated by anti-V region antibodies binding to TCRα and TCRβ within the same

TCR. Hence, the increase in FRET efficiency upon preincubation with tetramer could be produced by inter-TCR rather than intra-TCR changes within nanoclusters. To investigate inter-TCR changes produced by preincubation with tetramer in the absence of intra-TCR signals, we measured FRET signals using FRET donor and acceptor anti-Vβ5 antibodies conjugated to different fluorophores. We detected an increase in FRET efficiency with a maximum of 4 min that was completely abolished if TCR nanoclusters had been disrupted by MβCD treatment (Fig. 6e). These results indicate that binding of pMHC tetramer to the TCR produces changes in the quaternary structure of TCR nanoclusters. In fact, in T cells from double transgenic OT-1xHY mice, we could detect an increase in FRET intensity between two anti-Vα antibodies specific to the two different TCRs (Fig. 6f). The changes detected with anti-Vα and anti-Vβ antibodies, indicative of changes in the quaternary structure of the TCR, are coincident with the changes in the inter-tetramer ligand distance or orientation (Fig. 6b) and with changes in the cooperativity of tetramer binding (Fig. 3).

**Antigen presentation induces TCR cooperativity.** Once we determined that binding of a soluble pMHC multivalent ligand transiently favoured subsequent binding events, we next interrogated whether cooperativity upon ligand binding was also detected in a more physiological setting, namely in response to pMHC antigens presented by professional antigen-presenting cells (APCs). To this end, we incubated OT-1 CD8[+] T cells with mature dendritic cells (DCs) loaded with the OVA peptide antigen. The incubation was carried out at 37 °C to allow T cells to spread onto the APCs by a process that requires remodelling of the cytoskeleton[25]. In the time period studied, TCR down-regulation did not take place (Fig. 7a, left). However, we detected a clear cooperativity: preincubation with antigen-loaded DCs for 3–4 min improved binding of a soluble OVAp-PE tetramer, while longer incubations impaired it (Fig. 7a, centre). To avoid possible effects due to activation of intracellular signalling pathways by engaged TCRs, we pretreated OT-1 T cells with the Src tyrosine kinase inhibitor PP2 that is known to abrogate TCR signalling[26]. Although the intensity of tetramer binding to OT-1 cells treated with PP2 was reduced, the optimum time point of cooperative binding was similar (Fig. 7a, right). These results suggest that transient cooperative effects manifested upon ligand binding are also produced during the physiological activation of T cells.

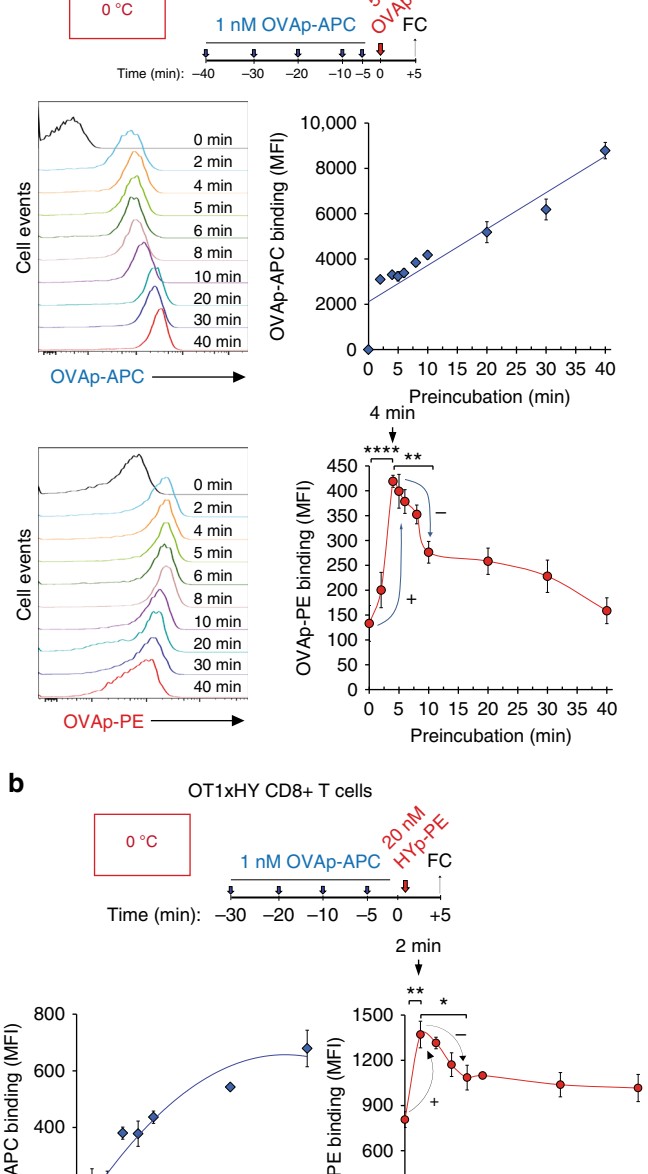

Fig. 3 Binding of a first pMHC tetramer ligand at sub-saturating concentrations transiently favours binding of a second pMHC tetramer ligand. **a** OT-1 T CD8 + T cells were preincubated at 0 °C with 1 nM OVAp-APC tetramer for the indicated times and subsequently incubated with 5 nM of OVAp-PE tetramer for five additional minutes. Time 0 indicates binding of OVAp-PE tetramer in the absence of preincubation with OVAp-APC tetramer. Cells were fixed and stained for CD8 before analysis by flow cytometry and calculation of the MFI for OVAp-PE and OVAp-APC. Histograms for OVAp-APC tetramers and OVAp-PE tetramer fluorescence intensity are shown on the left panels. The MFI values for OVAp-APC tetramers and OVAp-PE tetramer are represented on the right panels. **b** CD8 + T cells from double transgenic female OT-1xHY mice were preincubated on ice with 1 nM OVAp-APC tetramer for the indicated times and subsequently incubated with 20 nM of HYp-PE tetramer for five additional minutes. Cells were fixed and stained for CD8 before analysis by flow cytometry to generate MFI values for HYp-PE and OVAp-APC binding. All data shown in Fig. 3 represent the mean ± s.d. of triplicate datasets; *p < 0.05; **p < 0.005; ****p < 0.00005 (2-tail unpaired t-test)

High-resolution confocal microscopy imaging from previous studies showing preferential recruitment of effector proteins and phosphorylation of all TCRs in a restricted area of bigger microclusters, instead of an even distribution, are suggestive of cooperativity[27, 28]. We therefore looked at the distribution of OVAp tetramer binding, compared with that of the TCR at the T-cell–APC immunological synapse (IS). The location of a CD3ζ–GFP fusion protein was used as indication of the total

TCR distribution. OT-1 T cells were preincubated for 8 min with antigen-loaded DCs at 37 °C to allow the formation of an IS. Subsequently, we examined the sites for preferential pMHC binding in the IS by adding a soluble OVAp-APC tetramer for 5 min. We found that the OVAp tetramer bound only at the edges of the T-cell–DC contact site, whereas the centre-accumulated TCR that was not able to bind to OVAp tetramer (Fig. 7b). In contrast, the sites for more intense OVAp tetramer binding

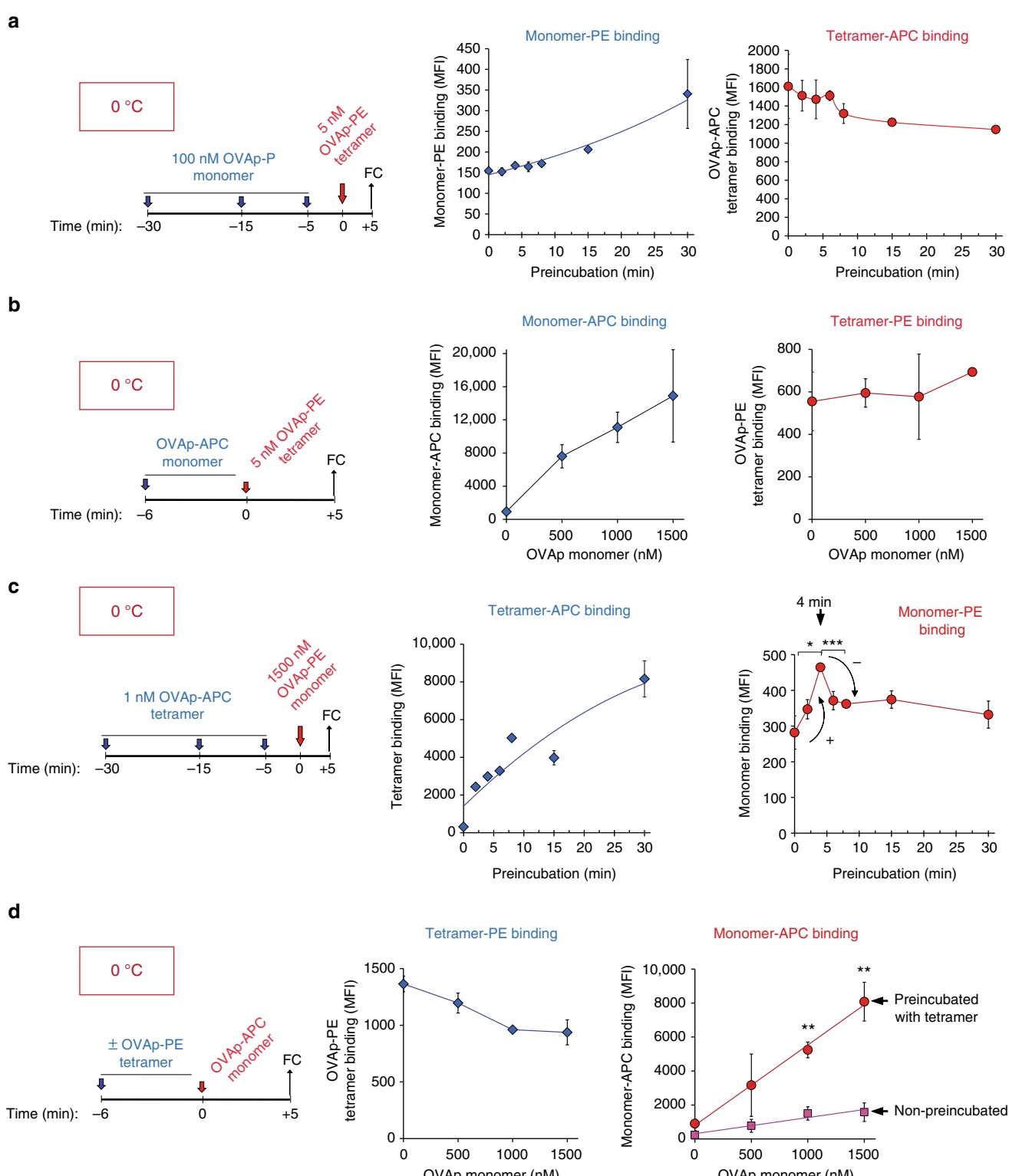

highly co-localised with the TCR in its Active conformation, as detected by APA1/1 staining, which was also present at the edges of the IS. The centre of the IS was stained with a large (>1000 kDa) extracellular anti-CD3 antibody/streptavidin complex (Supplementary Fig. 8), suggesting that poor binding of the OVAp tetramer is not due to inaccessibility of the reagent to the centre of the IS. Pearson correlation analysis of pixel distribution in different T-cell–DC conjugates identified a much better co-localisation of OVAp tetramer binding sites with APA1/1 than with total TCR (Fig. 7b). These results suggest that TCRs in the Active conformation are preferentially located at the periphery of the IS and that, consequently, it is where enhanced binding of soluble pMHC is detected.

## Discussion

In this paper, we show the existence of homotropic allostery in the TCR, i.e. the first-binding events of pMHC ligand to TCRs alter the binding properties of subsequent pMHC–TCR engagements. This is demonstrated by using pMHC oligomers of different nature and colours at sub-saturating concentrations. These first-binding pMHCs favour binding of a second pMHC oligomer to the vast majority of free TCRs still available. The cooperative effect could also be demonstrated by using T cells bearing two different TCR transgenes (OT-1 and HY) of unrelated pMHC specificity. We demonstrated that cooperative binding is possible because the TCR is organised in nanoclusters and our data therefore suggest that the OT-1 and HY TCRs are intermingled in the same nanoclusters. The positive cooperativity upon ligand binding manifested in the first 6–8 min after the initial binding events can be explained by the adoption of the Active conformation by the entire TCR nanocluster including the unbound TCRs at once. Consistent with the idea that Active conformation is required for positive cooperativity, OT-1 T cells expressing the C80G mutant of CD3ε did not show such cooperativity. We have previously shown that the C80G mutation in the stalk domain of CD3ε inhibits the outside-in transmission of conformational changes. Moreover, expression of the C80G mutant in T cells that express a large excess of endogenous wild-type CD3ε exerts a dominant negative effect on T-cell activation[29]. This early evidence of cooperativity was interpreted as a need for adoption of Active conformation by the entire TCR nanocluster. The existence of inter-TCR cooperativity has also been suggested in view of the non-random distribution of CD3ζ and ZAP70 phosphorylation hotspots at preferential sites of TCR clusters[27, 28]. However, these last data could also be explained by *trans*-acting effects of tyrosine kinases recruited to TCRs engaged by antigen. Our experiments of inter-TCR cooperativity manifested on pMHC ligand binding have been mostly carried out at 0 °C or in the presence of the Src tyrosine kinase inhibitor PP2 to minimise the participation of cytoplasmic effectors on the observed cooperative effects. The effect of disrupting TCR nanoclusters with MβCD, the correlation with optimum for APA1/1 binding, indicative of the Active conformation, the optimum for cooperative pMHC binding and the optimum for FRET intensity between TCRs all point to a rearrangement of the TCR quaternary structure as the basis for inter-TCR cooperativity. At present, we do not have evidence for the intermolecular mechanisms that can result in the transmission of the Active conformation from engaged to non-engaged TCRs within a nanocluster. Nevertheless, the enhanced FRET efficiency observed between two TCRβ or two TCRα subunits after preincubation with a pMHC tetramer suggest that movement of the αβ TCR ectodomains is ultimately the consequence of lateral spreading of the Active conformation from engaged to non-engaged TCRs. Indeed, enhanced binding of monomeric pMHC promoted by preincubation with a pMHC tetramer suggests that the pMHC-binding distal tips of the αβ ectodomains move away from the plasma membrane pointing towards the extracellular milieu as a way to facilitate pMHC binding.

In an attempt to investigate if inter-TCR cooperativity upon ligand binding is also manifested during antigen recognition in physiological conditions, i.e. antigen presented by antigen-presenting cells, we have also carried out flow cytometry experiments, in which we can detect a sharp temporal increase in soluble pMHC binding when T cells are stimulated with antigen-loaded DCs. This phenomenon was also corroborated by confocal microscopy experiments. High-resolution microscopy would be needed to pinpoint the locations at which the TCR is in the Active conformation within the IS. Nonetheless, with the methods employed here, we see a correlation between the TCR in the Active conformation, detected with APA1/1, and the sites for enhanced pMHC tetramer binding. We also show that the centre of the IS is enriched in TCR, but not in enhanced pMHC binding. A possible explanation for the poor staining of the IS centre is size exclusion: a tight apposition of the DC and T-cell membranes at the centre may preclude the diffusion of the pMHC tetramer towards the centre. We show, however, that a large complex of tetrameric anti-CD3 antibody is able to enter and stain the centre of the IS. These data argue against a size-exclusion mechanism. We rather propose that the centre of the IS is enriched in TCRs in the Inhibited conformation. According to this model, TCR allostery is manifested at the periphery of the IS, whereas at the centre of the IS, the TCR presumably accumulates in its Inhibited conformation. These data are in line with previous confocal microscopy data showing that the TCR is associated at the periphery of the IS with intracellular signalling proteins, whereas it is

**Fig. 4** A multimeric pMHC ligand is required to promote cooperative pMHC binding. **a** CD8 + OT-1 T cells were preincubated at 0 °C with 100 nM biotinylated OVAp monomer for the indicated times and subsequently incubated with 5 nM of OVAp-APC tetramer for five additional minutes. Time 0 indicates binding of OVAp-APC tetramer in the absence of preincubation with OVAp monomer. Cells were fixed and stained with streptavidin-PE in order to detect binding of the monomer. Plots represent the MFI values for OVAp-PE monomer and OVAp-APC tetramer binding. **b** CD8 + OT-1 T cells were incubated at 0 °C with the indicated concentrations of biotinylated OVAp monomer for 6 min and subsequently incubated with 5 nM of OVAp-PE tetramer for 5 additional minutes. Cells were then fixed and stained with streptavidin-APC in order to detect the monomer. Plots represent the MFI values for OVAp-APC monomer and OVAp-PE tetramer binding. **c** CD8 + OT-1 T cells were preincubated at 0 °C with 1 nM OVAp-APC tetramer for the indicated times and subsequently incubated with 1.5 µM of biotinylated OVAp monomer (detected with streptavidin-PE) for 5 additional minutes. Time 0 indicates binding of OVAp monomer in the absence of preincubation with OVAp-APC tetramer. Cells were fixed and stained with streptavidin-PE in order to detect binding of the monomer. Plots represent the MFI values for OVAp-PE monomer and OVAp-APC tetramer binding. **d** CD8 + OT-1 T cells were preincubated or not at 0 °C with 1 nM of OVAp-PE tetramer for 6 min prior to incubation with the indicated concentrations of biotinylated OVAp monomer for five additional minutes. Cells were fixed and stained with streptavidin-APC in order to detect the monomer. Plots represent the MFI values for OVAp-APC monomer and OVAp-PE tetramer binding. All data shown in Fig. 4 represent the mean ± s.d. of triplicate datasets; * $p < 0.05$; ** $p < 0.005$; *** $p < 0.0005$ (2-tail unpaired $t$-test)

essentially devoid of those effectors at the centre[25, 30, 31]. In fact, it has been previously been proposed that the TCR accumulates at the centre of the IS in a signalling-incompetent manner and from there it is internalised and removed[30, 32, 33]. The signalling incompetence of the TCR at the centre together with the large concentration of TCR and MHC in that location could be explained by our data that support the idea of TCR adopting the Inhibited conformation at the centre. The translocation time for engaged TCRs from the periphery to the centre of the IS is roughly coincident with the time window for the TCR to undertake the Resting to→Active to→Inhibited transitions[34, 35]. At the periphery of the IS, the first engagements

of the TCR favour engagement of subsequent ones in the same nanocluster, resulting in signal amplification. The requirements for pMHC affinity of the second set of engaged TCRs might be lesser than those of the first. Thus, signal amplification may be achieved during the secondary binding events by involving MHC complexes loaded with low-affinity ligands, such as self-peptides[19, 36]. Adoption of the Inhibited conformation and loss of cooperativity might be envisioned as a mechanism for T cells to count the abundance of antigen because sustained signalling can only be achieved through the recruitment of new TCRs from other sites of the plasma membrane or from cytoplasmic trafficking vesicles.

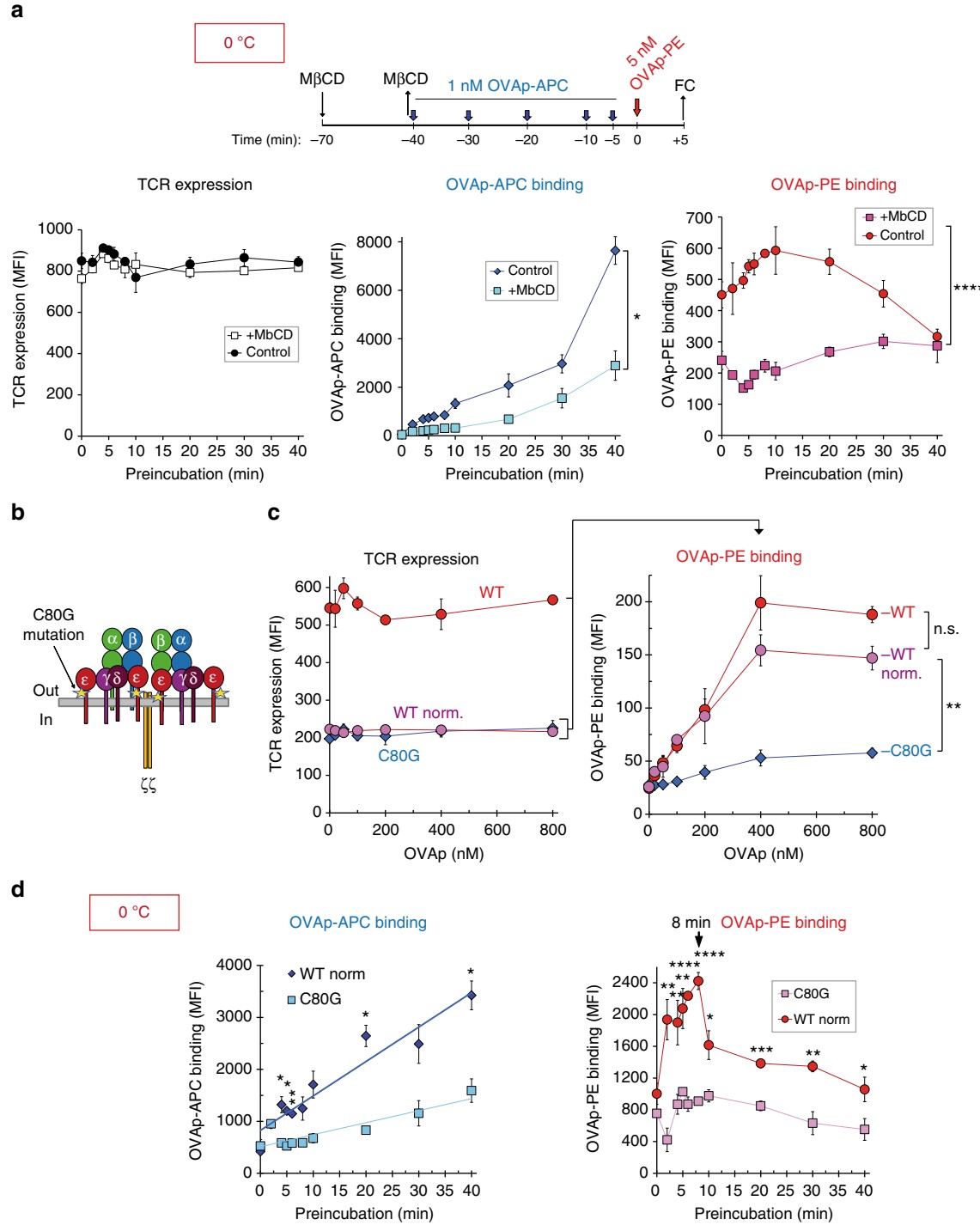

## Methods

**Mice**. OT-1 mice express a transgenic TCR H-2K$^b$ restricted and specific for ovalbumin peptide SIINFEKL (OVAp)[37]. The OT-1 CD3ζ-GFP transgenic mice express a wild-type CD3ζ subunit fused with the green fluorescent protein in a OT-1 CD3ζ$^{-/-}$ background[38]. The OT-1 C80G knock-in mice express a CD3ε subunit with a cysteine-to-glycine mutation in the stalk sequence of the ectodomain[22]. The HY TCR is H-2D$^b$ restricted and specific for the male HY antigen[39]. T cells from female HY TCR transgenic mice were used for all experiments involving this line. All the mice were maintained under specific pathogen-free conditions at the animal facility of the CBMSO in accordance with the current national and European guidelines. All animal procedures were approved by the ethical committee of the CSIC.

**Cell preparation**. Murine naïve OT-1 T cells were obtained from lymph nodes from 6- to 8-week-old female mice by homogenisation and washing in phosphate buffered saline (PBS) containing 10% (vol/vol) foetal bovine serum (FBS). T lymphoblasts were obtained by homogenisation of the spleen, washing in PBS containing 10% (vol/vol) FBS and stimulating splenic T cells with OVAp for two days, then washing and expanding them with recombinant interleukin 2 (IL-2) for another four days in RPMI medium [RPMI-10 supplemented with 0.01% sodium pyruvate and 10 μM β-mercaptoethanol][40]. For dendritic cell differentiation, bone marrow precursors were obtained from femurs of 6–12-week-old C57BL/6 mice and grown for 10–11 days of differentiation with GM--CSF, as previously described[41].

**pMHC tetramers and other reagents**. Allophycocyanin (APC)-conjugated OVAp/H-2K$^b$ (SIINEFKL), phycoerythrin (PE)-conjugated OVAp/H-2K$^b$, unlabelled OVAp/H-2K$^b$ and PE-conjugated HY2p/H-2D$^b$ tetramers, and biotinylated OVAp/H-2K$^b$ monomers were obtained from TCMetrix. OVAp mutant for CD8 binding (OVAp CD8-blocked/H-2K$^b$, Q226D227K-OVAp) tetramer was produced as previously described[42]. APC-conjugated OVAp/H-2K$^b$ (SIINEFKL) and PE-conjugated OVAp/H-2K$^b$dextramer were obtained from Immudex. The following antibodies were obtained from BD Pharmingen and used at the following concentrations: fluorescein isothiocyanate (FITC)-conjugated anti-CD8α (1:100), peridinin chlorophyll protein complex (PerCP)-conjugated anti-CD3 (2C11) (1:50), PE-conjugated anti-CD3 (2C11) (1:50, 1:25, 1:10), APC-conjugated anti-CD3 (17A2) (1:10) and APC-conjugated anti CD27 (1:50). Brilliant Violet 421-conjugated anti CD8α (1:100) was obtained from BioLegend. Antibody staining was performed in PBS, 1% bovine serum albumin (BSA) and 0.02% sodium azide during 30 min on ice. Methyl-beta cyclodextrin (MβCD) was obtained from Sigma-Aldrich and used at a final concentration of 10 mM to treat cells for 30 min in RPMI medium at room temperature. PP2 was obtained from Calbiochem and used at a final concentration of 20 μM for 30 min at 37 °C.

**Sequential binding of pMHC tetramers and APA1/1 staining**. All the experiments were performed in triplicates. For each condition, a total of $2 \times 10^5$ naïve T cells or T lymphoblasts diluted in 30 μl of RPMI medium, buffered with 10 mM Hepes, pH = 7.4, were stimulated with different concentrations of tetramers, monomers or dextramers sequentially diluted in 10 μl of PBS, 1% BSA and 0.02% sodium azide during different incubation times. Sodium azide was included in all incubations on ice, but not at 37 °C to avoid metabolic side effects. When we performed sequential incubations using two different tetramers or one tetramer and a biotinylated monomer, tetramers were preincubated with an excess of free biotin (1 mM) in order to block all possible free-biotin binding sites from the streptavidin used for tetramerization and to prevent cross-labelling. Tetramer binding was stopped by a fourfold dilution with ice-cold 2% paraformaldehyde (PFA), rapid spinning, washing with PBS, 1% BSA and 0.02% sodium azide and further fixation in 2% PFA for 20 min on ice. The fixed cells were then stained with antibodies against the appropriate extracellular markers or

streptavidin-APC for monomer detection and, where indicated, after permeabilization with PBS, 1% BSA, 0.02% sodium azide and 0.01% saponin, cells were stained with the APA1/1 monoclonal antibody (10 μg/ml). Alexa Fluor 488-conjugated anti-mouse antibody was used to visualise APA1/1 staining[22]. For experiments using a blocking anti-CD8 antibody, cells were incubated previously with a saturating concentration of an anti-CD8-blocker (CT-CD8a Thermo Scientific) for 40 min at 0 °C. For the cooperativity analysis using dendritic cells, bone marrow dendritic cells (DCs) were differentiated during 10 days and loaded with OVAp for 2–3 h before stimulation. Subsequently, a total of $1 \times 10^5$ OVAp-loaded DCs were plated onto 96-well plates, and $2 \times 10^5$ naïve OT-1 T cells (ratio 1:2) were added in a final volume of 20 μl. The DC + T-cell culture mixtures were incubated for different times at 37 °C and, subsequently, cells were incubated with 5 nM of APC-conjugated OVAp/H-2K$^b$ tetramer for 5 min, and then fixed and stained as described above.

**Confocal microscopy and immunofluorescence**. For each condition, a total of $1 \times 10^5$ bone-marrow-derived DCs were directly differentiated on glass coverslips during 10 days, and loaded with OVAp for 2–3 h before T-cell stimulation. A total of $2 \times 10^5$ purified lymph node CD8$^+$ T cells from OT-1-CD3ζ-GFP mice were added onto the DCs and incubated at 37 °C for different times. Subsequently, APC-conjugated H-2k$^b$ tetramer at a final concentration of 20 nM either alone or in combination with a biotinylated anti-CD3ε (145-2C11) pre-conjugated with streptavidin-Alexa 555 was added for 5 additional min, and they were then fixed and permeabilized as described previously[43]. Staining with the APA1/1 monoclonal antibody was done after cell permeabilization, and Alexa Fluor 555-conjugated anti-mouse was used as secondary antibody. Cells were examined on a Zeiss Confocal LSM510 META microscope coupled to a vertical Axiovert200 microscopy system with a 63 × Plan-Apochromat M27 oil immersion objective lens (1.4 numerical aperture). For quantification, 16-bit images were analysed using Fiji software[44]. Briefly, the intensity on the plasma membrane was quantified in cells at the regions of interest (ROIs), and the MFI of each region was measured for each channel. Where indicated, quantitative co-localisation was carried out using the Pearson Index intensity correlation analysis.

**FRET analysis**. For FRET analysis between TCR chains, a total of $2 \times 10^5$ OT-1 or OT-1xHY lymph node naïve T cells were resuspended in complete RPMI medium and incubated on ice with 1 nM of unlabeled H-2Kb-OVAp tetramer for different times. Cells were fixed with 2% PFA and subsequently stained at saturating conditions for 30 min at 0 °C using different combinations of anti-TCR chains conjugated to APC (acceptor) or PE (donor) in PBS, 1% BSA and 0.02% sodium azide. For FRET analysis between tetramers, a total of 200.000 OT-1 naïve T cells were resuspended in complete RPMI medium and incubated on ice with 1 nM of PE-conjugated H-2Kb-OVAp and 1 nM of APC-conjugated H-2Kb-OVAp tetramers in the presence of free biotin for different times, and they were fixed with 2% PFA. The cells were analysed using a BD FACS Canto II equipped with violet (405 nm), blue (488 nm) and red (633 nm) lasers. Using the blue laser line, PE and FRET emissions were collected using 585/42 and LP670 filters, respectively, while after delayed excitation with the red laser line, APC emission was collected using a 660/20 filter. For each combination of antibodies, single controls for acceptor and donor were included in order to correct their contribution to other channels. A negative control using the acceptor and an anti-CD27-APC was also included for all antibody combinations. Mean fluorescence intensity (MFI) for each channel was calculated for the double positive cells (PE$^+$ APC$^+$), or simple positive cells for the controls, using FlowJo software. FRET efficiency was calculated using MFI for each channel as previously described[45].

**Size-exclusion chromatography and fraction analysis**. Sephacryl S-300 (GE-Healthcare) was slurried and washed twice with Hepes buffered saline (HBS) and subsequently degassed and packed in a 6-mL column 16/10. The column void

---

**Fig. 5** Ligand binding cooperativity depends on TCR nanoclusters adopting the Active conformation. **a** OT-1 T cells were incubated with 10 mM methyl-β-cyclodextrin (MβCD) or vehicle for 30 min at 37 °C, cooled to 0 °C and preincubated with 1 nM OVAp-APC tetramer for the indicated times, then incubated with 5 nM OVAp-PE for 5 additional min. Cells were fixed and stained with the anti-TCRβ antibody H57 to assess TCR levels. Left panel: TCR expression at each time point of preincubation with OVAp-APC for non-treated and MβCD-treated cells. Middle panel: OVAp-APC tetramer binding at different times of preincubation. Right panel: OVAp-PE tetramer binding during 5 min as a function of preincubation time with the OVAp-APC tetramer. **b** Cartoon of a TCR indicating the position of the Cys80 residue mutated to glycine in the extracellular domain of CD3ε. **c** CD8 + OT-1 T cells expressing wild type or C80G mutant CD3ε were incubated at 0 °C with the indicated concentrations of OVAp-PE tetramer for 5 min. Cells were fixed and stained with the anti-CD3 antibody 2C11 to assess TCR expression (left panel). Tetramer binding data (right panel) was obtained from WT and CD3ε C80G cell populations, and from a WT subpopulation gated for TCR expression similar to CD3ε C80G cells (WT$_{norm}$, left panel). **d** CD8 + OT-1 T cells expressing wild type or C80G CD3ε were preincubated at 0 °C with 1 nM OVAp-APC tetramer for the indicated times and with 5 nM OVAp-PE for 5 additional minutes. Cells were fixed and stained with the anti-TCRβ antibody H57 to normalise TCR levels. Left panel: OVAp-APC tetramer binding at different times of preincubation. Right panel: OVAp-PE tetramer binding during 5 min for all preincubation times in WT C80G OT-1 cells. OVAp-APC and OVAp-PE tetramer binding is compared between CD3ε C80G mutant cells and CD3ε WT cells normalised for equal TCR expression as in **c**. All data shown in Fig. 5 represent the mean ± s.d. of triplicate datasets; *$p < 0.05$; **$p < 0.005$; ***$p < 0.0005$; ****$p < 0.00005$ (2-tail unpaired t-test)

volume (Vo) was calculated from the elution volume for Blue Dextran 2000, and calibration of the column was calculated from the elution volume of proteins with known molecular weights. After calibration, a 60-μl sample of OVAp-PE tetramer was loaded onto the column followed by elution with HBS. Fraction collection

started immediately after the column void volume had passed. The eluate was collected in 96 sequential fractions of 75 μl. To evaluate the composition of each fraction, 10 μl of each fraction was incubated for 40 min with $2 \times 10^5$ OT-1 CD8 + cells and analysed by flow cytometry. In addition, 10 μl of each fraction was

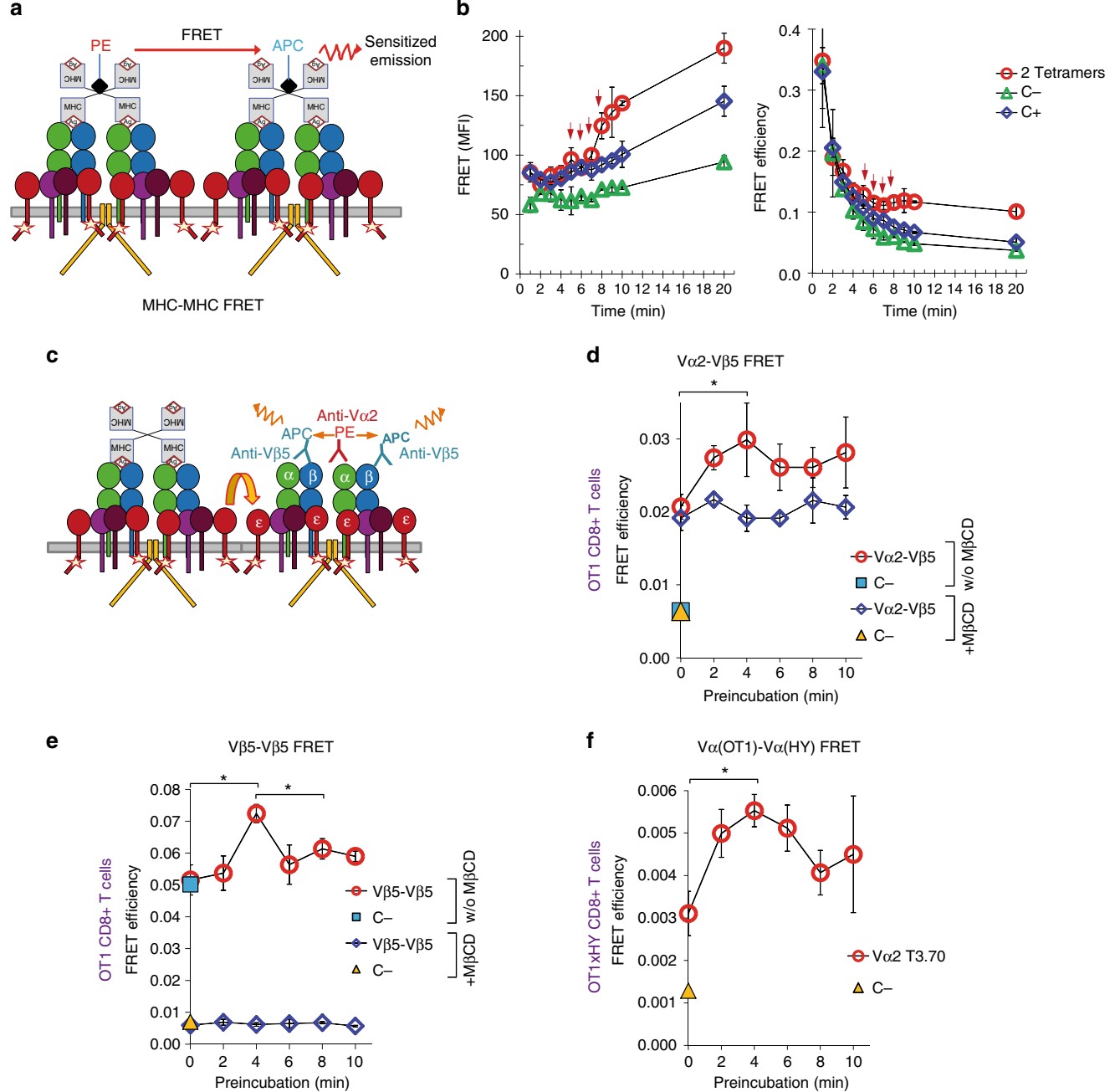

**Fig. 6** Ligand binding induces changes in the proximity or orientation of TCRs within nanoclusters. **a** Schematic of the experimental set-up of Fig. 6b. FRET changes were detected upon simultaneous binding of PE- and APC-labelled OVAp tetramers. **b** CD8 + OT-1 T cells were incubated on ice with 5 nM each of OVAp-PE and OVAp-APC tetramers for the indicated times and fixed. FRET intensity and efficiency (left and right panels) were measured by flow cytometry. For FRET controls, OT-1 T cells were incubated with 5 nM of the OVAp-PE tetramer for the indicated times and, after fixation, cells were further incubated for 50 min with either APC-labelled anti-CD3 (positive control, C + ) or APC-labelled anti-CD27 (negative control, C-) at saturating concentrations. Arrows indicate the time points at which an upward tendency of the FRET slope was noticed. **c** Schematic of the experimental set-up in **d-f**. FRET changes were detected in CD8 + OT-1 T cells that bound PE-labelled OVAp tetramer and APC-labelled anti-Vβ5 antibody. **d-f** CD8 + OT-1 or OT-1xHY T cells pretreated or not with 10 mM MβCD for 30 min at 37 °C and subsequently were incubated at 0 °C with 1 nM unlabelled OVAp tetramer for the indicated times. After fixation, they were incubated for 50 min with anti-Vα2-PE and anti-Vβ5-APC (**d**), anti-Vβ5-PE and anti-Vβ5-APC (**e**) or anti-T3.70-PE and Vα2-APC (**f**). For all antibody combinations, a negative control is included using the donor antibody and anti-CD27-APC (C-). FRET efficiency is calculated as described in the Methods section and represented versus time of preincubation with tetramer. All data shown in Fig. 6 represent the mean ± s.d. of triplicate datasets; *p < 0.05 (2-tail unpaired t-test)

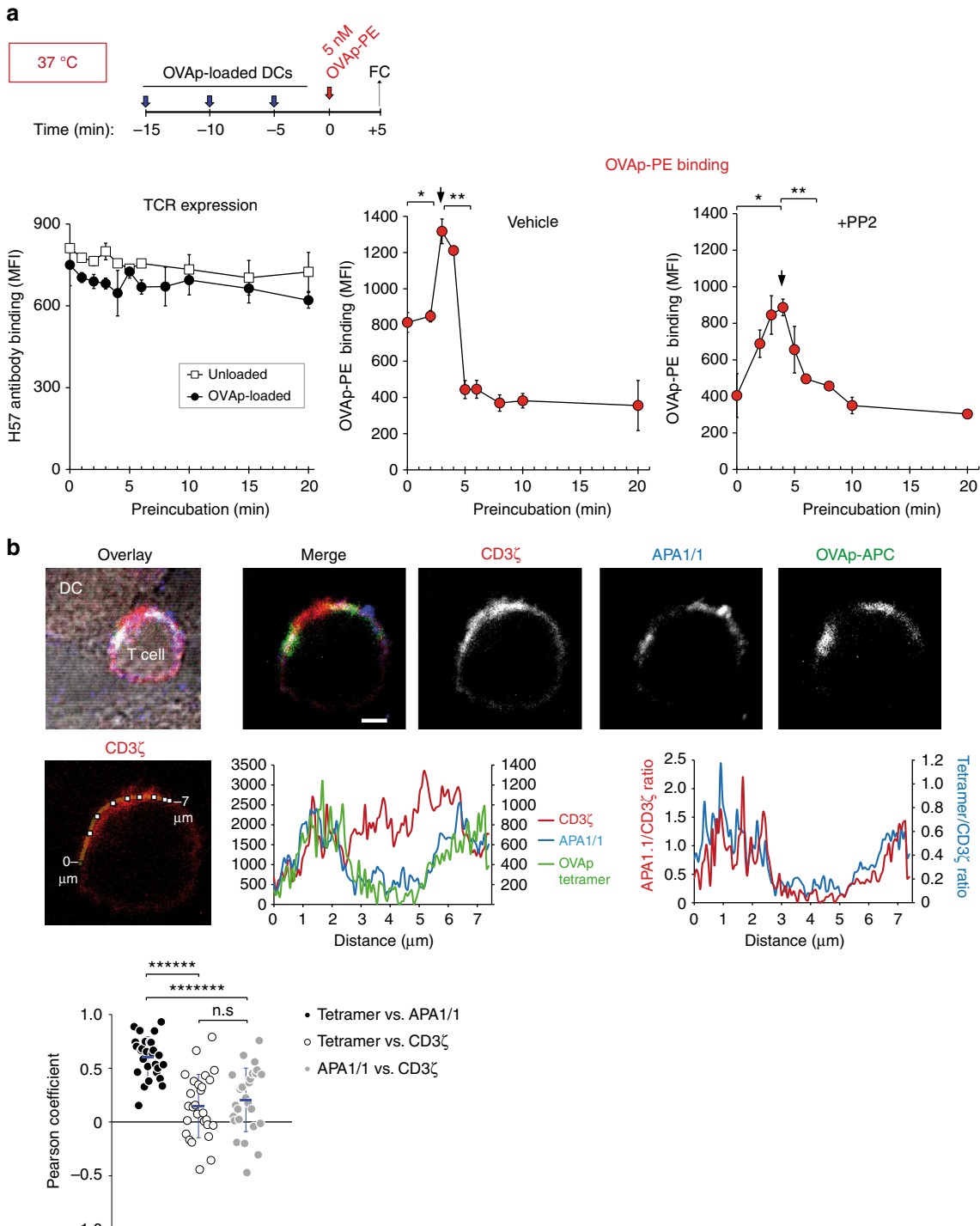

**Fig. 7** The sites of enhanced tetramer binding co-localise with the Active TCR at the periphery of the immunological synapse. **a** CD8 + OT-1 T cells were treated with 20 μM PP2 or vehicle for 30 min at 37 °C and then preincubated for the indicates times at 37 °C in the presence of mouse bone-marrow-derived dendritic cells (mDCs) loaded with OVA peptide. Subsequently, cells were incubated with 5 nM OVAp-PE for 5 min. Afterwards, cells were fixed and stained with anti-CD8 and anti-H57 antibodies to quantify TCR expression. **b** CD8 + T cells from OT-1 CD3ζ-GFP mice were purified by negative selection and incubated for 8 min at 37 °C in the presence of mDCs attached to a coverslip and pre-loaded with OVA peptide. Subsequently, cells were incubated with 20 nM OVAp-APC for 5 min. Before visualisation by confocal microscopy, cells were fixed, permeabilized and stained with APA1-1 (top panel). Scale bar represents 1 μm. Line scan drawn along the synaptic zone to show relative fluorescence intensity of CD3ζ, APA1-1 and OVAp-tetramer (middle left panel). Relative fluorescence intensities of CD3ζ, APA1-1 and OVAp-tetramer are represented versus distance (middle centre). Ratios between the relative fluorescence intensity of APA1-1 and CD3ζ or OVAp-tetramer and CD3ζ are represented versus distance (middle right). Pearson correlation coefficients between the tetramer and APA1-1, the tetramer and CD3ζ, or APA1-1 and CD3ζ intensity distributions (bottom panel) are for $n = 27$ immunological synapses

resolved by SDS–PAGE and western blotting with biotin-labelled horseradish peroxidase.

**Statistical analysis**. Data are reported as means ± SD of multiple individual experiments each carried out in triplicate. Unless stated otherwise, the statistical analysis was carried out with GraphPad Prism 6.0. A two-tailed $t$-test was used if two groups were compared. Differences were considered significant if $P$ values were less than 0.05.

**Data availability**. The authors declare that the data supporting the findings of this study are available within the paper and its Supplementary Information files, and are available from the author upon request.

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

## Acknowledgements

We are indebted to Cristina Prieto, Valentina Blanco and Tania Gómez for their expert technical assistance and José Faro for helpful comments on the FRET experiments and mathematical modelling. This work was supported by the grant SAF2013-47975-R (to B.A.) from the CICYT, grant from the European Research Council ERC 2013-Advanced Grant 334763 'NOVARIPP' (to B.A.), the EU through grant FP7/2007-2013 (SYBILLA, to E.P., W.W.S. and B.A.), the Excellence Initiative of the German Research Foundation (DFG) EXC294 (Center for Biological Signalling Studies, BIOSS, to W.W.S.) through grants FIS2013-47949-C2-2-P, FIS2016-78883-C2-2-P and PIRSES-GA-2012-317893 (to M.C., C.M-P. and G.L.) and the Fundación Ramón Areces (to the CBMSO).

## Author contributions

N.M-B., R.B., E.R.B. and C.A. carried out all biological experiments; B.A. supervised the research; M.C., G.L. and C. M-P. built the mathematical model; B.A., M.C. and C.M-P.

prepared the manuscript which was revised by N.M-B., R.B., C.A., C.L.O., E.P., W.W.S. and G.L.. Statistical analysis was performed by B.A.

## Additional information

**Competing interests:** The authors declare no competing interests.

