## [Peer Review File · Nature Communications]

Reviewers' comments:

Reviewer #1 (Remarks to the Author):

Alarcon and co-workers have been the first proponents of a conformational model to explain TCR signal transduction. A key evidence supporting this model was the increased detection by the antibody APA/1 of a determinant in the intracellular tail of the CD3e subunits, after the TCR was triggered. Later, the same authors suggested that a geometrically-constrained clustering of TCRs was necessary to trigger such a conformational change and intracellular signaling. However, it still remains unclear whether peptide-MHC and/or the TCR are oligomers in the steady state and how such clustering leads to the next crucial event of phosphorylation-mediated signal propagation.

In this work, Martin-Blanco et al. have investigated in further detail the behaviour of the TCR-CD3 complex upon reacting TCR transgenic primary T cells with an agonist peptide-MHC rendered tetramer by streptavidin, an artificial polyvalence that supposedly induces receptor clustering and results in measurable intracellular signaling.

Here, the authors have used ultra-low (one-digit nM) concentration of peptide-MHC tetramers that barely allows detect binding to cells by flow-cytometry (Suppl. Fig 1a). The intended reason for this experimental set up is not explained, but it may be an attempt to observe potential effects of limited cluster formation on the binding and signalling property of the clustered TCRs or it originated from a serendipitous observation that at low ligand binding the authors observed an apparently unexpected result and decided to investigate this further.

The disadvantage of this setting is that the entire work relies on very small numbers above background, at the limits of signal-to-noise (see below). Moreover, while low ligand-to-TCR stoichiometry should favour high valence of interaction for a tetramer (relatively high proportion of TCR clustered as trimers), it may suffer of poor mass action, thus reducing the desired effect. Under these experimental conditions, weak APA/1 reactivity increase was observed. It would have been desirable to present as supplementary data a dose-response curve of tetramer ligand vs. APA/1 binding (by double fluorescence plot, but after background subtraction, see below) at early time (e.g. 4-5 minutes) to understand how APA/1 reactivity quantitatively relates to ligand binding. The authors observed that peptide-MHC binding increases in time, apparently not plateauing even after 20 minutes. In contrast, APA/1 reactivity increase ~ 3 fold over baseline level in about 5-6 minutes, in agreement with increased peptide-MHC binding, and declines

afterwards to a level that remains persistently slightly above or at background level. At face value, this indicates that the conformational change is transient and drastically reduced even though the ligand is bound or continue to bind. The disappearance of the conformational change is not explained or investigated in this work and remains difficult to rationalise in a simple allosteric model. The binding trend of the tetramer and the detection of the CD3e determinant decline are equally observed at 0 and 37 °C and not due to ITAM phosphorylation hindering APA/1 binding.

I feel that the work has potential for unearthing new facets of TCR signalling but I have serious concerns about the experimental set up and the lack of essential controls. These doubts should be exhaustively dissipated before considering molecular explanations and modelling.

1 - First, peptide-MHC tetramer binding engages so few TCRs that the signal-to-noise ratio must be very low, so that the evaluation of specific MFI becomes problematic. The only way to obtain a more reliable determination of a specific MFI is to measure the background for both peptide-MHC tetramers and APA/1 binding. Background data could be obtained in principle using the Ova tetramer on another TCR tg T cell or by inhibiting Ova tetramer binding with Fab fragments of anti-Vb and/or Va Ab that should not elicit signalling or cross-linking.

2 - I suggest to rigorously verifying the molecular dispersion, by a simple gel filtration, of the tetramers used. There have been several claims in the literature that preparations of peptide-MHC tetramers often contain sensible proportions of aggregated material, with consequent misvaluation of the actual binding to T cells and its physical meaning and, possibly, increasing background.

3 - A close inspection of the binding plots for peptide-MHC tetramers shown in figure 1 (and all the others) reveal a fast and slow component. This is reminiscent of the presence of specific (fast kinetic component) and non-specific (slow kinetic component) binding. It may be that, when non-specific binding is assessed (see above) and subtracted from the binding curve values, one could obtain a resulting binding curve that saturates at about the same time point when the detection of the APA/1 epitope is maximal. So, the “extra-binding” may not be due to more “activated” TCRs available but simply to the linear increase of background with time. The decline in the APA/1 antibody reactivity must also be re-evaluated after APA/1 background subtraction. The decline could be in part due to some exceeding steric hindrance caused by clustering or to other more complex chemical or physical causes. Thus, the binding measures must be carefully re-examined to verify if the mathematical modelling and the interpretation of the phenomena observed make sense.

4 - This also applies to the claim that there is essentially no difference in binding between 0 and 37 °C. For such a difference of temperature there must be differences of K_{on} and K_{off} of

tetramers binding. Also, the physical status of the plasma membrane changes substantially between these temperatures with possible consequences on TCR aggregation status and detection of the conformational changes. This is not because conformational changes cannot happen at low temperatures as well, but simply because the membrane lipid behave differently. The reason for not observing differences at 0 and 37 °C, could be that the sensitivity of the assay is not good enough also in the face of background that was not subtracted.

5 - Other data in this work, including the imaging and the FRET need also a substantial revision. It is unclear how FRET efficiency due to two contiguous TCRs could be higher than FRET efficiency between two probes binding to the same molecule (the positive control). Finally the confocal images are in my opinion, over-interpreted. They simply show at best, that engaged receptors show increased detection of APA1 epitope.

6 - Conceptually, I see serious problems in the rationalization of the model. The authors should be explicit in clarifying the consequences of their model on T cell activation.

In order to explain their data and their interpretation through the lens by the mathematical model, the authors assume the existence of a TCR “resting state”, defined by them as a physical configuration unable to bind the peptide-MHC. Then, T cells activation must occur only through the fraction of pre-existing “active” TCRs, in a configuration able to bind Ova-MHC tetramers. It is unclear what proportion the “active” TCR must be present at steady state in order to ensure rapid activation of the T cell, commensurate to ligand abundance and affinity. Are “active” TCRs in the steady state those already in clusters? Why paradoxically TCRs reactivity needs to be kept at bay? Is it a “trick” to ensure ligand discrimination? These questions need to be discussed. Instead, we are left with the counter-intuitive idea that the number of “activated TCR” is fixed a priori and could be even very low (different in naive vs. non-activated T cells?) and that the extent of activation will be pre-determined by that number, no matter how many agonist ligands are presented on an APC, thus imposing apparent restrictions on T cell functional fitness.

Reviewer #3 (Remarks to the Author):

The mechanism underlying T cell response to pathogen-derived peptides presented by MHCs is a long-standing puzzle in the field of immunology. Preclustering of TCRs on a T cell surface and cooperativity between different TCRs as a result of conformational changes in response to initial antigen binding were previously theoretically suggested as a mechanism by which T cell might enhance antigen recognition. Manuscript by Martin-Blanco et al. proposed a novel assay to quantify that cooperativity using a new readout of binding of monoclonal antibody APA1/1 that detects a conformational change in the cytoplasmic tail of CD3e. Their main findings could be

summarized as: (1) preincubation of OT1-T cells with pMHCs tetramers was shown to lead to an upregulation of APA1/1 antibody binding but only in the narrow window of preincubation times; (2) surprisingly, similar effect was also observed using CD8 T cells from double transgenic mice co-expressing two different TCRs (OT-1 and HY); (3) the effect did not depend on blocking of CD8 co-receptor, but the cholesterol extraction with methyl- β -cyclodextrin abrogated it; (4) using FRET they showed that shift in the proximity of TCR-bound tetramers occurred at the point within the observed narrow time window for upregulation of APA1/1 antibody binding; (5) using confocal microscopy they showed that enhanced tetramer binding colocalized with the area of enhanced APA1/1 antibody binding; and, finally, (6) they proposed a three-state model for TCR transition that suggest a mechanism to explain the data and is consistent with their experimental results.

These are novel results experimentally showing a cooperativity between T cell receptors in response to initial antigen binding and providing mechanism to explain the observed results. The results will be of great interest to the T cell immunology field.

Major points to address:

In the abstract “By disrupting nanoclusters and preventing the transmission of conformational changes, we demonstrate that pMHC-binding cooperativity requires TCRs in nanoclusters in their Active state”. I would soften that conclusion using “we demonstrate that observed pMHC-binding cooperativity is consistent with the model of TCRs in nanoclusters...” instead of “requires”.

One of the main results of the paper is the observation of the maximum at the exposure of the APA1/1 epitope in the narrow time window of preincubation of the T cells with pMHCs tetramers. The maximum time slightly varies between different figures (e. g. Fig. 1b and Fig. 2e). On the line 86, authors said that in the model “This maximum is independent of different parameters, most importantly ligand density and the temperature”. What model parameters do determine the time of the maximum for the number of nanoclusters in the Active state? How sensitive this maximum time to these parameters?

To illustrate the point that ligand-mediated TCR cross-linking is required for the observed cooperativity effect, the authors showed extended Figure 3b-d. I am confused with that figure for the following reasons. First, I feel like the key figure is missing as there is no figure similar to Figure 3d for the case when cells first preincubated with monomers and after that with tetramers. Second, tetramer concentration used is 1nM while the monomer concentrations used for comparison are two order of magnitude higher. Will the authors explain their choice of the concentrations? Third, when compare Figure 3b and 3c, showing the opposite cases, you can see that x-axes are the same on all panels. So, my question is why not to use a symmetrical experimental design? If I understand correctly, authors would like to illustrate that preincubation

with tetramers lead to maximum in monomer binding while preincubation with monomers does not lead to peak in tetramer binding.

Figure 2e. The tetramer binding is reduced for the case of C80G mutation, but shape-wise the blue diamond curve looks similar to red circle curve. Both curves show an increase following with a decrease and even have similar slopes on the declining phase. So, I am confused with conclusion made that “C80G mutation prevented the transient enhancement in OVAp-PE tetramer binding”. I would say that effect is reduced instead of prevented.

Line 113: This is a very curious interesting result. Does it suggest that OT1 and HY TCRs are in the same pre-formed cluster?

Figure 1e. Why triangles showing the inhibited state start not from zero on y-axis? Do you expect some nanoclusters initially in inhibited state?

Did authors consider to apply the same approach for weaker OT-1 T cell ligands to see if cooperativity effect still hold? Of course, this could be a subject for another study.

Minor points:

line 45: suggestion to remove “(Fig.1b)” as it is confusing at this place. Instead I would put “(Fig.1a)” at the end of the second sentence in line 46 which will also bring the referring to the figures in order.

line 52: correct the sentence “When binding at a sub-saturating...” -> “When tetramers binding to T cells at a sub-saturating...”

line 81: suggestion to replace word “appearance” with word “status” or “transition between”. If I understand correctly nanoclusters are preset in the model, they do not appear.

line 371: main title to figure 1 (in bold) is confusing

line 382: typo in the sentence

Table 3 refers to non-existing Sec. 1.4 and Sec 1.5

lines 96-99: sentence is not clear. I would strongly recommend someone with English as a first language to look at the manuscript text.

line 104: the word “fixed” seems unnecessary

line 105: on the referred figure, I see “sharp raise” not “sharp peak”

line 106: should it be 4min not 5min (see corresponding figure)?

line 159: typo “to”->”with”

line 184: typo “in”->”as”

Point-by point reply

Reviewer #1 (Remarks to the Author):

Alarcon and co-workers have been the first proponents of a conformational model to explain TCR signal transduction. A key evidence supporting this model was the increased detection by the antibody APA/1 of a determinant in the intracellular tail of the CD3 ϵ subunits, after the TCR was triggered. Later, the same authors suggested that a geometrically-constrained clustering of TCRs was necessary to trigger such a conformational change and intracellular signaling. However, it still remains unclear whether peptide-MHC and/or the TCR are oligomers in the steady state and how such clustering leads to the next crucial event of phosphorylation-mediated signal propagation.

The referee is absolutely right in that our new data strongly support a model in which the TCR is pre-clustered forming oligomers called nanoclusters *before* the TCR has been bound by any peptide-MHC and that it is in this pre-existing oligomeric organization in which TCR crosslinking by pMHC takes place and the TCR is triggered. We and others have contributed with different data to support the existence of TCR nanoclusters before engagement by pMHC:

- 1) Biochemical techniques.-
 - Ultracentrifugation in density gradients (Exley et al., 1995)
 - Co-IP of 2 different TCRs (Fernandez-Miguel et al., 1999).
 - Electrophoresis under native conditions (Schamel et al., 2005)
- 2) Electron microscopy.-
 - Label-fracture and visualization of the “apical” T cell membrane (Kumar et al., 2011; Schamel et al., 2005)
 - Intracellular staining of the “basal” T cell membrane (Lillemeier et al., 2009)
- 3) Confocal microscopy.-
 - Single molecule resolution (Lillemeier et al., 2009; Sherman et al., 2011)
 - Quantum dot distribution (Zhong et al., 2009)
- 4) FRET and spectroscopy.-
 - Cross-correlation spectroscopy (James et al., 2007)
 - FRET between 2 different TCRs using antibodies (Fernandez-Miguel et al., 1999) and *this paper*
 - FRET between 2 pMHC tetramers: *This paper*
- 5) Genetic evidence.-
 - Disruption of nanocluster with a single point mutation (Kumar et al., 2011)
- 6) Other techniques.-
 - Non-random distribution of pMHC binding events (Zarnitsyna et al., 2007)

In addition, there is evidence supporting the idea that MHC loaded with agonistic peptide antigen of the same kind is organized in pre-existing clusters *before* the antigen-presenting cells establish contact with cognate T cells. This evidence has been found for APCs loaded with soluble antigen peptide and also for APCs infected with a virus:

- MHC-I nanoclusters loaded with cognate antigen peptide in virus-infected cells (Ferez et al., 2014)
- MHC-II nanoclusters expressing the same cognate antigen on the surface of DCs (Bosch et al., 2013)

All this idea of preformed TCR nanoclusters as the framework for signal transmission is in fact supported by the present work not only because we find a quite important FRET signal between two bound tetramers (at 0°C) but also from the main message of the paper: there is inter-TCR cooperativity manifested on ligand binding properties that requires the TCR to be pre-organized in nanoclusters.

In this work, Martin-Blanco et al. have investigated in further detail the behaviour of the TCR-CD3 complex upon reacting TCR transgenic primary T cells with an agonist peptide-MHC rendered tetramer by streptavidin, an artificial polyvalence that supposedly induces receptor clustering and results in measurable intracellular signaling.

OVAp tetramers have previously been shown to trigger T cell activation (Cochran et al., 2000) and we use them here as suitable reagents to quantify ligand binding to the TCR. In addition, we show that the main result can also be reproduced with dextran-based pMHC oligomers and when using peptides presented by APCs. This demonstrates that the phenomenon we describe, namely the inter-TCR cooperativity upon ligand binding, is also manifested using physiological ways of antigen presentation. In addition, we now show that low nanomolar concentrations of the OVAp tetramers we use throughout the manuscript are able to activate OT1 T cells as determined by CD69 expression (Fig. 1b) and by CD25 expression (Extended Data Fig. 1a). We also show that the artificial soluble OVAp tetramer is able to induce TCR clustering above the size of the pre-existing nanoclusters (if cells are stimulated at 37°C) by confocal and electron microscopy (Extended Data Fig. 6). These results show that the artificially polyvalent pMHC tetramers used are able to activate T cells. The need for pMHC polyvalence for T cell activation was previously described (Boniface et al., 1998; Cochran et al., 2000). However, we also show that inter-TCR cooperativity on ligand binding is also manifested using physiological ways of antigen presentation (Fig. 7a) which indicates that the phenomenon we describe is not only detected using an artificial OVAp tetramer (or dextramer) stimulus.

Here, the authors have used ultra-low (one-digit nM) concentration of peptide-MHC tetramers that barely allows detect binding to cells by flow-cytometry (Suppl. Fig 1a). The intended reason for this experimental set up is not explained, but it may be an attempt to observe potential effects of limited cluster formation on the binding and signalling property of the clustered TCRs or it originated from a serendipitous observation that at low ligand binding the authors observed an apparently unexpected result and decided to investigate this further. We apologize for the unclear justification of the use of one-digit nanomolar concentrations of pMHC tetramers for our experiments in the original version of the manuscript. We used such low concentrations of pMHC tetramers, because we aimed to be far below binding saturation, i.e. in conditions at which most of the TCRs are free. This was done on purpose, in order to study cooperativity on ligand binding and we needed most of the TCR after incubation with pMHC tetramers to be free. In this way, we were able to measure subsequent binding events. The need to have most of the TCR free to study cooperativity was not important when using T cells expressing two TCRs of different specificity (OT1xHY; Fig. 3b) but it was very important for the experiments using two tetramers in different colours for a single specificity T cell (OT1).

Nonetheless, we have now included experiments of T cell activation (CD69 and CD25 expression) by a wide range of pMHC tetramer concentrations in which we find that the maximal response is reached at these low concentrations, i.e. 1-10 nanomolar (Fig. 1b and Extended Fig. 1a). We think that the reasoning behind using low nanomolar concentrations in our experiments is justified by both the maximal T cell response and the need to be far below binding saturation. We hope that with the new data and the changes introduced in the Text we have made this point much clearer.

The disadvantage of this setting is that the entire work relies on very small numbers above background, at the limits of signal-to-noise (see below). Moreover, while low ligand-to-TCR stoichiometry should favour high valence of interaction for a tetramer (relatively high proportion of TCR clustered as trimers), it may suffer of poor mass action, thus reducing the desired effect. Under these experimental conditions, weak APA/1 reactivity increase was observed. It would have been desirable to present as supplementary data a dose-response curve of tetramer ligand vs. APA/1 binding (by double fluorescence plot, but after

background subtraction, see below) at early time (e.g. 4-5 minutes) to understand how APA/1 reactivity quantitatively relates to ligand binding. The authors observed that peptide-MHC binding increases in time, apparently not plateauing even after 20 minutes. In contrast, APA/1 reactivity increase ~ 3 fold over baseline level in about 5-6 minutes, in agreement with increased peptide-MHC binding, and declines afterwards to a level that remains persistently slightly above or at background level. At face value, this indicates that the conformational change is transient and drastically reduced even though the ligand is bound or continue to bind. The disappearance of the conformational change is not explained or investigated in this work and remains difficult to rationalise in a simple allosteric model. The binding trend of the tetramer and the detection of the CD3e determinant decline are equally observed at 0 and 37 °C and not due to ITAM phosphorylation hindering APA/1 binding.

We have followed the recommendation of Referee #1 to illustrate the concentration dependence for increased abundance of the *Active* conformation (as seen by APA/1 staining). We have used a very low concentration (1 nM) and another 200-fold higher to show that the maximum time point of APA/1 epitope exposure (~6 min) does not change by increasing the concentration of the pMHC tetramer ligand (Fig. 1d). Unlike the 1 nM condition, with the 200 nM concentration a plateau of OVAp tetramer binding is reached at 6-8 min. Nevertheless the MFI of APA/1 binding decreases to pre-activation levels after 10 min. We think that this new data adds further support to the idea expressed in the mathematical model (and Fig. 1b) that it is not necessary to saturate the TCR to get a maximal response, because all TCRs in a nanocluster adopt the *Active* conformation at once, even if most of the TCRs in the nanocluster are not bound by the ligand.

I feel that the work has potential for unearthing new facets of TCR signalling but I have serious concerns about the experimental set up and the lack of essential controls. These doubts should be exhaustively dissipated before considering molecular explanations and modelling.

We have carried out new experiments (those in Fig. 1b, 1d, 1f and Extended Data Fig. 1a, 1b, 1c, 1d) and introduced additional controls (staining HY TCR-bearing CD8+ T cells, staining of CD4+ T cells, gel filtration) to show that changes in APA/1 staining and tetramer binding are significantly above background and are not artefactual. We hope with the new set of data to have dissipated the doubts that the referee mentioned.

1 - First, peptide-MHC tetramer binding engages so few TCRs that the signal-to-noise ratio must be very low, so that the evaluation of specific MFI becomes problematic. The only way to obtain a more reliable determination of a specific MFI is to measure the background for both peptide-MHC tetramers and APA/1 binding. Background data could be obtained in principle using the Ova tetramer on another TCR tg T cell or by inhibiting Ova tetramer binding with Fab fragments of anti-Vb and/or Va Ab that should not elicit signalling or cross-linking.

We understand the concern of Referee #1 though we are completely sure that both tetramer and APA/1 binding MFI values are specific and above background. As the referee recommended, we have used another transgenic TCR mouse line (HY) of irrelevant specificity as a negative control for both OVAp tetramer and APA/1 binding. We show in Extended Data Fig. 1b that OVAp tetramer binds to CD8+ OT1 T cells with values higher than those of binding to HY CD8+ T cells. Furthermore, while binding to OT1 cells increases with incubation time, the MFI for HY remains level. Likewise, APA/1 MFI increased transiently with incubation, reaching a maximum at ~ 5 min in OT-1 T cells, whereas it remained uniform in HY T cells. Furthermore, we show flow cytometry plots in Fig. 1f and Fig. 3a that illustrate the increase in fluorescence intensity both for APA/1 and OVAp tetramer stainings, a difference that can even be seen with the naked eye.

2 - I suggest to rigorously verifying the molecular dispersion, by a simple gel filtration, of the tetramers used. There have been several claims in the literature that preparations of peptide-MHC tetramers often contain sensible proportions of aggregated material, with consequent

misvaluation of the actual binding to T cells and its physical meaning and, possibly, increasing background.

We have verified the molecular dispersion of our tetramers by gel filtration as the referee suggested and found no binding capacity for OT-1+ CD8+ T cells in molecular weights above 500 kDa (Extended Data Fig. 1d) in line with the idea that we have tetramers (considering the molecular weight of the 4 MHC-I heavy chains, the 4 beta2-microglobulin molecules and the molecular weight of the streptavidin-phycoerythrin moiety). As a control of binding specificity we show the MFI values for binding of the different fractions to CD4+ T cells in the same samples. There is no binding to the latter cells. Furthermore, we do not detect MHC-I heavy chain by western blot in fractions corresponding to a molecular weight > 669 kDa. Overall, we do not think that the changes in binding to OT-1 CD8+ T cells with time are due to the presence of aggregates in the tetramer preparations.

3 - A close inspection of the binding plots for peptide-MHC tetramers shown in figure 1 (and all the others) reveal a fast and slow component. This is reminiscent of the presence of specific (fast kinetic component) and non-specific (slow kinetic component) binding. It may be that, when non-specific binding is assessed (see above) and subtracted from the binding curve values, one could obtain a resulting binding curve that saturates at about the same time point when the detection of the APA/1 epitope is maximal. So, the “extra-binding” may not be due to more “activated” TCRs available but simply to the linear increase of background with time. The decline in the APA/1 antibody reactivity must also be re-evaluated after APA/1 background subtraction. The decline could be in part due to some exceeding steric hindrance caused by clustering or to other more complex chemical or physical causes. Thus, the binding measures must be carefully re-examined to verify if the mathematically modelling and the interpretation of the phenomena observed make sense.

As mentioned above, binding of APA1/1 and OVAp tetramers to OT-1 CD8+ T cells is clearly above background, defined as binding to irrelevant HY CD8+ T cells or to CD4+ T cells. In addition, we do not agree with the interpretation of “fast” and “slow” kinetic components. The shape of the binding curves displays a non-linear growing trend similar to that in Fig. 2e but with the unavoidable experimental fluctuations. Moreover, as shown in Fig. 1c, 1 nM is far from binding saturation (even 200nM is) so the alternative explanation posed by the reviewer is not sustained by the data.

4 - This also applies to the claim that there is essentially no difference in binding between 0 and 37 °C. For such a difference of temperature there must be differences of K_{on} and K_{off} of tetramers binding. Also, the physical status of the plasma membrane changes substantially between these temperatures with possible consequences on TCR aggregation status and detection of the conformational changes. This is not because conformational changes cannot happen at low temperatures as well, but simply because the membrane lipid behave differently. The reason for not observing differences at 0 and 37 °C, could be that the sensitivity of the assay is not good enough also in the face of background that was not subtracted.

According to Alam et al. 1999, *Immunity*, 10, 227-37, there are no drastic differences in the on and off rates of the pMHC-TCR interaction:

$k_{off}=0.028$ 1/sec and $k_{on}=2040$ 1/(M·sec) at 37°C and

$k_{off}=0.022$ 1/sec and $k_{on}=3720$ 1/(M·sec) at 25°C

Similarly, Rossette et al. 2001, *Immunity*, 15, 59-70, found:

$k_{off}=0.00632$ 1/sec and $k_{on}=4720$ 1/(M·sec) at 0°C

We agree with the Referee that biophysical parameters concerning the fluidity and arrangement of the membrane might strongly depend on the temperature. However, our data indicate that those changes do not strongly impact on the cooperativity phenomena we describe here. As we have shown by new control experiments, the background is not an issue (see above).

The predictions of the mathematical model are identical for all sets of parameters. So, precisely because the rates are different (especially in comparison with 0°C), is what makes the entropic interpretation of the landscape in Extended Data Fig. 1c (and consequently, the model built upon that) more robust: the effect strong changes in binding due to the temperature are not reflected in the changes in the relative proportion of clusters in different states (active, resting and inhibited).

5 - Other data in this work, including the imaging and the FRET need also a substantial revision. It is unclear how FRET efficiency due to two contiguous TCRs could be higher than FRET efficiency between two probes binding to the same molecule (the positive control). Finally the confocal images are in my opinion, over-interpreted. They simply show at best, that engaged receptors show increased detection of APA1 epitope.

-We were also surprised by the higher FRET efficiency found between 2 pMHC tetramers compared to the positive control (tetramer and anti-CD3) since the 2 tetramers need to bind different TCRs whereas the anti-CD3 antibody can bind to the same TCR to which the pMHC tetramer is bound. The FRET efficiency depends on the proximity of the probes and on their orientation. This has been demonstrated for instance using FRET pairs bound to dsDNA of defined length and structure (Iqbal et al., 2008). A possible explanation to the higher FRET efficiency between 2 tetramers is that they will bind the 6 CDR loops of the TCR α/β ectodomains, thus most likely being oriented in a way that the PE and APC fluorochromes are in a FRET-favourable orientation. On the other hand, the anti-CD3 antibodies (including 2C11, used in this paper) are known to bind CD3 ϵ in a diagonal orientation; away from the pMHC binding site at the tips of the TCR α/β ectodomains (Fernandes et al., 2012; Kjer-Nielsen et al., 2004). Thus, we suggest that both, proximity and orientation, might play a role in detecting higher FRET between two pMHC tetramers compared to the one between pMHC tetramer and anti-CD3 antibodies.

-The confocal microscopy data of Figure 7b and Extended Figure 8 supports the main result of Figure 7a indicating that physiological antigen presentation does also induce cooperative binding. As the referee suggests, it is possible that it is the addition of tetramer during the last 5 minutes before fixation what induces the exposure of APA1/1. However, this interpretation would not explain why the center of the immunological synapse, which contains the higher concentration of TCR (detected with anti-CD3z), is not stained with APA1/1. It could be argued that APA1/1 does not stain the center of the synapse because the tetramer is unable to reach that space due to size restrictions. However, we show in Extended Figure 8 that a probe bigger than the MHCp tetramer, formed by tetrameric anti-CD3, is able to access the center of the synapse and stain it. Therefore, we think that it is reasonable to propose that the TCR is first engaged by antigen presented by DCs and that later it becomes prone or not to bind tetramer, such as we have shown in the previous figures with two soluble tetramers. Besides, we have done additional confocal microscopy work to show that APA1/1 stains the periphery of the synapse, even if tetramer is not added. We have not included that additional control because we think it does not offer new information, but we would be happy to share the data with the referee, if requested.

6 - Conceptually, I see serious problems in the rationalization of the model. The authors should be explicit in clarifying the consequences of their model on T cell activation.

In order to explain their data and their interpretation through the lens by the mathematical model, the authors assume the existence of a TCR “resting state”, defined by them as a physical configuration unable to bind the peptide-MHC. Then, T cells activation must occur only through the fraction of pre-existing “active” TCRs, in a configuration able to bind Ova-MHC tetramers. It is unclear what proportion the “active” TCR must be present at steady state in order to ensure rapid activation of the T cell, commensurate to ligand abundance and affinity. Are “active” TCRs in the steady state those already in clusters? Why paradoxically TCRs reactivity needs to be kept at bay? Is it a “trick” to ensure ligand discrimination? These

questions need to be discussed. Instead, we are left with the counter-intuitive idea that the number of “activated TCR” is fixed a priori and could be even very low (different in naive vs. non-activated T cells?) and that the extent of activation will be pre-determined by that number, no matter how many agonist ligands are presented on an APC, thus imposing apparent restrictions on T cell functional fitness.

We apologize that we did not make this point clearer in the earlier version of the manuscript. We do not claim that the *Resting* state is unable to bind to pMHC. Our data rather suggest that the *Active* TCR conformation enhances pMHC tetramer binding.

As shown in Fig. 1e, the fraction of clusters in the *Active* state in non-engaged T cells is negligible and it is the enhancement of the *Active* state due to cross-linking that creates signal amplification. This creates a window of opportunity for TCR signalling by increasing the sensitivity.

This conclusion does not follow from either our data or our model. The amount of agonist ligand is important as long as it enhances cross-linking. This means that the "functional" element of the TCR is the cluster rather than the individual receptor. This is sustained by the fact that soluble monomeric MHCp is not capable of activating the TCR, according to Refs: (Abastado et al., 1995; Boniface et al., 1998; Cochran et al., 2000; Minguet et al., 2007)

References

- Abastado, J.P., Lone, Y.C., Casrouge, A., Boulot, G., and Kourilsky, P. (1995). Dimerization of soluble major histocompatibility complex-peptide complexes is sufficient for activation of T cell hybridoma and induction of unresponsiveness. *J Exp Med* 182, 439-447.
- Boniface, J.J., Rabinowitz, J.D., Wulfing, C., Hampl, J., Reich, Z., Altman, J.D., Kantor, R.M., Beeson, C., McConnell, H.M., and Davis, M.M. (1998). Initiation of signal transduction through the T cell receptor requires the multivalent engagement of peptide/MHC ligands [corrected]. *Immunity* 9, 459-466.
- Bosch, B., Heipertz, E.L., Drake, J.R., and Roche, P.A. (2013). Major histocompatibility complex (MHC) class II-peptide complexes arrive at the plasma membrane in cholesterol-rich microclusters. *J Biol Chem* 288, 13236-13242. doi: 13210.11074/jbc.M13112.442640. Epub 442013 Mar 442626.
- Cochran, J.R., Cameron, T.O., and Stern, L.J. (2000). The relationship of MHC-peptide binding and T cell activation probed using chemically defined MHC class II oligomers. *Immunity* 12, 241-250.
- Exley, M., Wileman, T., Mueller, B., and Terhorst, C. (1995). Evidence for multivalent structure of T-cell antigen receptor complex. *Mol Immunol* 32, 829-839.
- Ferez, M., Castro, M., Alarcon, B., and van Santen, H.M. (2014). Cognate peptide-MHC complexes are expressed as tightly apposed nanoclusters in virus-infected cells to allow TCR crosslinking. *J Immunol* 192, 52-58. doi: 10.4049/jimmunol.1301224. Epub 1302013 Dec 1301224.
- Fernandes, R.A., Shore, D.A., Vuong, M.T., Yu, C., Zhu, X., Pereira-Lopes, S., Brouwer, H., Fennelly, J.A., Jessup, C.M., Evans, E.J., et al. (2012). T cell receptors are structures capable of initiating signaling in the absence of large conformational rearrangements. *J Biol Chem* 287, 13324-13335. doi: 13310.11074/jbc.M13111.332783. Epub 332012 Jan 332719.
- Fernandez-Miguel, G., Alarcon, B., Iglesias, A., Bluethmann, H., Alvarez-Mon, M., Sanz, E., and de la Hera, A. (1999). Multivalent structure of an alphabeta T cell receptor. *Proc Natl Acad Sci U S A* 96, 1547-1552.
- Iqbal, A., Arslan, S., Okumus, B., Wilson, T.J., Giraud, G., Norman, D.G., Ha, T., and Lilley, D.M. (2008). Orientation dependence in fluorescent energy transfer between Cy3 and Cy5 terminally attached to double-stranded nucleic acids. *Proc Natl Acad Sci U S A* 105, 11176-11181. doi: 11110.11073/pnas.0801707105. Epub 0801702008 Aug 0801707101.

James, J.R., White, S.S., Clarke, R.W., Johansen, A.M., Dunne, P.D., Sleep, D.L., Fitzgerald, W.J., Davis, S.J., and Klenerman, D. (2007). Single-molecule level analysis of the subunit composition of the T cell receptor on live T cells. *Proc Natl Acad Sci U S A* *104*, 17662-17667. Epub 12007 Oct 17630.

Kjer-Nielsen, L., Dunstone, M.A., Kostenko, L., Ely, L.K., Beddoe, T., Mifsud, N.A., Purcell, A.W., Brooks, A.G., McCluskey, J., and Rossjohn, J. (2004). Crystal structure of the human T cell receptor CD3 epsilon gamma heterodimer complexed to the therapeutic mAb OKT3. *Proc Natl Acad Sci U S A* *101*, 7675-7680. Epub 2004 May 7610.

Kumar, R., Ferez, M., Swamy, M., Arechaga, I., Rejas, M.T., Valpuesta, J.M., Schamel, W.W., Alarcon, B., and van Santen, H.M. (2011). Increased Sensitivity of Antigen-Experienced T Cells through the Enrichment of Oligomeric T Cell Receptor Complexes. *Immunity* *35*, 375-387. Epub 2011 Sep 2018.

Lillemeier, B.F., Mortelmaier, M.A., Forstner, M.B., Huppa, J.B., Groves, J.T., and Davis, M.M. (2009). TCR and Lat are expressed on separate protein islands on T cell membranes and concatenate during activation. *Nat Immunol* *11*, 90-96.

Minguet, S., Swamy, M., Alarcon, B., Luescher, I.F., and Schamel, W.W. (2007). Full activation of the T cell receptor requires both clustering and conformational changes at CD3. *Immunity* *26*, 43-54. Epub 2006 Dec 2021.

Schamel, W.W., Arechaga, I., Risueno, R.M., van Santen, H.M., Cabezas, P., Risco, C., Valpuesta, J.M., and Alarcon, B. (2005). Coexistence of multivalent and monovalent TCRs explains high sensitivity and wide range of response. *J Exp Med* *202*, 493-503. Epub 2005 Aug 2008.

Sherman, E., Barr, V., Manley, S., Patterson, G., Balagopalan, L., Akpan, I., Regan, C.K., Merrill, R.K., Sommers, C.L., Lippincott-Schwartz, J., *et al.* (2011). Functional nanoscale organization of signaling molecules downstream of the T cell antigen receptor. *Immunity* *35*, 705-720. Epub 2011 Nov 2014.

Zarnitsyna, V.I., Huang, J., Zhang, F., Chien, Y.H., Leckband, D., and Zhu, C. (2007). Memory in receptor-ligand-mediated cell adhesion. *Proc Natl Acad Sci U S A* *104*, 18037-18042. Epub 12007 Nov 18038.

Zhong, L., Zeng, G., Lu, X., Wang, R.C., Gong, G., Yan, L., Huang, D., and Chen, Z.W. (2009). NSOM/QD-based direct visualization of CD3-induced and CD28-enhanced nanospatial coclustering of TCR and coreceptor in nanodomains in T cell activation. *PLoS ONE* *4*, e5945.

Reviewer #3 (Remarks to the Author):

Major points to address:

In the abstract “By disrupting nanoclusters and preventing the transmission of conformational changes, we demonstrate that pMHC-binding cooperativity requires TCRs in nanoclusters in their Active state”. I would soften that conclusion using “we demonstrate that observed pMHC-binding cooperativity is consistent with the model of TCRs in nanoclusters...” instead of “requires”.

We thank the referee for this kind advice and have followed the suggestion. Taking the opportunity to remodel the sentence, it now reads: “we demonstrate that pMHC-binding cooperativity is consistent with the model of TCRs being in nanoclusters that coordinately adopt the *Active* state”. We think that the sentence is now softer and also better reflects what we are proposing.

One of the main results of the paper is the observation of the maximum at the exposure of the APA1/1 epitope in the narrow time window of preincubation of the T cells with pMHCs tetramers. The maximum time slightly varies between different figures (e. g. Fig. 1b and Fig. 2e). On the line 86, authors said that in the model “This maximum is independent of different parameters, most importantly ligand density and the temperature”. What model parameters do determine the time of the maximum for the number of nanoclusters in the Active state? How sensitive this maximum time to these parameters?

We agree with the reviewer about the strength of the claim. We have changed that sentence accordingly to: "The location of this maximum is insensitive to parameter changes, such as ligand concentration, temperature and free-energy differences between the three states (Methods and Extended Data Fig. 2b)." Note that Extended Data Fig 2b shows how the location of the maximum is within a window (like the inter-experiment variability) displayed as a grey horizontal "band" for a wide range of parameters. This is the robustness or insensitivity we are alluding to. Other parameters (not shown in that figure) are obtained from the literature or estimated as discussed in the Extended Data Section "Mathematical model and parameter values".

To illustrate the point that ligand-mediated TCR cross-linking is required for the observed cooperativity effect, the authors showed extended Figure 3b-d. I am confused with that figure for the following reasons. First, I feel like the key figure is missing as there is no figure similar to Figure 3d for the case when cells first preincubated with monomers and after that with tetramers. Second, tetramer concentration used is 1nM while the monomer concentrations used for comparison are two order of magnitude higher. Will the authors explain their choice of the concentrations? Third, when compare Figure 3b and 3c, showing the opposite cases, you can see that x-axes are the same on all panels. So, my question is why not to use a symmetrical experimental design? If I understand correctly, authors would like to illustrate that preincubation with tetramers lead to maximum in monomer binding while preincubation with monomers does not lead to peak in tetramer binding.

Following Referee #3's advice, we have carried out such an experiment that is now displayed in Figure 4a. We see no significant effect of preincubation with the monomer on binding of the tetramer either in a time-dependent or concentration-dependent manner (Fig. 4a and 4b), whereas preincubation with the tetramer favours, in a transient manner, binding of the monomer (Fig. 4c and 4d). We now think that the Figure is complete and show it as a main Figure.

Figure 2e. The tetramer binding is reduced for the case of C80G mutation, but shape-wise the blue diamond curve looks similar to red circle curve. Both curves show an increase following with a decrease and even have similar slopes on the declining phase. So, I am confused with conclusion made that “C80G mutation prevented the transient enhancement in OVAp-PE tetramer binding”. I would say that effect is reduced instead of prevented.

We have now used “strongly inhibited” instead of “prevented” to leave open the possibility that some upregulation in the C80G mutant is possible

Line 113: This is a very curious interesting result. Does it suggest that OT1 and HY TCRs are in the same pre-formed cluster?

We thank the reviewer for this keen observation. Indeed, that is exactly our interpretation of the cooperativity data between OT-1 and HY TCRs (Fig. 3b and Extended Data Fig 3b) and also of the FRET data between the TCR α chain of the OT-1 TCR and the TCR α chain of the HY TCR (Fig. 6f).

Figure 1e. Why triangles showing the inhibited state start not from zero on y-axis? Do you expect some nanoclusters initially in inhibited state?

That is correct. This is a reflection of the landscape depicted in Extended Data Fig1c. The diagram reflects the relative fraction of allosteric ensembles within each conformation. Besides, the fact that the response is temperature-independent supports the idea that it is the number of possible “degrees of freedom” characterized by each state which determines the depth of each well.

Note that, while we are interested in the events after the addition of ligand, the cluster is a dynamic entity with a myriad of short-lived changes giving rise to the mentioned landscape.

Did authors consider to apply the same approach for weaker OT-1 T cell ligands to see if cooperativity effect still hold? Of course, this could be a subject for another study.

We have begun to use H-2Kb tetramers loaded with peptides of weaker affinity for the OT1 TCR as the Referee suggests. We are using tetramers just above or below the negative selection threshold (Q4R7 and Q4H7). We have found that we can have a time optimum for APA1/1 epitope exposure of 6 min using 20 nM of Q4R7 but a delayed time optimum of 10 min by using an optimal concentration of 100 nM of Q4H7. In terms of cooperativity in binding, we have confirmed, as predicted, that preincubation with the strong agonist (OVAp tetramer) favours the binding of the weaker agonists (Q4R7 and Q4H7) but we still need more time to see if 20 nM of Q4R7 and 100 nM of Q4H7 do or do not induce cooperative binding of OVAp tetramer. As the referee suggests, a full study will be necessary to investigate the effect of affinity on cooperativity induction. This will be especially interesting if there is a shift in the time optimum with below-threshold ligands, but we are not sure of this at present.

Minor points:

line 45: suggestion to remove “(Fig.1b)” as it is confusing at this place. Instead I would put “(Fig.1a)” at the end of the second sentence in line 46 which will also bring the referring to the figures in order.

Following the Referee’s advice, we have reformatted the Figures in the paper and put the different panels of Figure 1 in order.

line 52: correct the sentence “When binding at a sub-saturating...” -> “When tetramers binding to T cells at a sub-saturating...”

We have now corrected that awkward sentence.

line 81: suggestion to replace word “appearance” with word “status” or “transition between”. If I understand correctly nanoclusters are preset in the model, they do not appear.

Done. We have used “Transition of nanoclusters between...”

line 371: main title to figure 1 (in bold) is confusing

We have now replaced the title with the following one: “TCRs are in equilibrium between three different states of conformations” which we think is less confusing.

line 382: typo in the sentence

We thank the referee for noticing the typo. We have now replaced “of” by “or”.

Table 3 refers to non-existing Sec. 1.4 and Sec 1.5

We have now corrected the Table.

lines 96-99: sentence is not clear. I would strongly recommend someone with English as a first language to look at the manuscript text.

We have tried to clarify the sentence as much as possible and the Text has been corrected by a native speaker. In fact, CLO and GL are both native speakers.

line 104: the word “fixed” seems unnecessary

Done

line 105: on the referred figure, I see “sharp raise” not “sharp peak”

We have replaced the word as the referee suggested.

line 106: should it be 4min not 5min (see corresponding figure)?

We have changed the sentence to explain better the results in Extended Data Figure 3. We now wrote: “We found a sharp raise in binding of the second tetramer upon 4 min of preincubation that slowly decayed from 4 to 6 min of preincubation and decayed faster from 6 min onwards”.

line 159: typo “to”->”with”

Done

line 184: typo “in”->”as”

Done

Reviewers' comments:

Reviewer #1 (Remarks to the Author):

There continue to be incoherence in the binding data and flow cytometry and lack of adequate controls.

1 - A main issue in figure 1 is the odd result that binding to cells of pMHC tetramers at 200 nM (Fig 1d) exposes the epitope of (APA/1), supposed to detect only the TCR intracellular tail, at a value (~ 150 MFI) lower than the the that attained by 1 nM (200 times lower ligand concentration!). Even if the authors are correct that there is a “lateral spreading” of TCR activation when just one is activated, using 200 times more ligand, one should see that many more supposed pre-existing clusters are activated. How to explain (for example at 2 min stimulation) that 1 nM tetramer gives a 10 MFI for binding with an MFI for APA/1 similar/reduced as compared to 200 nM tetramer giving an MFI for pMHC binding of 600 (60 times higher)?

2 - “Spreading” of conformational changes in a cluster should take milliseconds/seconds. But the data presented indicate that APA/1 epitope access reaches a maximum only at 6 minutes. There are suggestions of such mechanisms for membrane ion channels that indicate these lateral phenomena must be much faster. Besides, how can one explain that tyrosine phosphorylation of the engaged TCR for a given amount of ligand usually takes seconds to be detected and peaks at 1 minute, the appearance of the APA/1 epitope peaks only after 6 minutes?

3 - I had suggested to perform tetramer binding on control T cells. The authors did it but instead of using 1 nM pMHC tetramer they used 5 nM (figure 1f, with no numbers in the x-axes). Why so if they need to control for the data in figure 1d that depicts experiments carried at 1 nM? Because the numbers are at times so small (10 to 30) that one cannot see real differences. Besides, the binding data in figure 1f are meaningless without adequate statistics. There are ways to improve flow-cytometry data by labelling control cells with viable fluorescence dye and perform analyses at the same time on control samples. Examples of background variability and incongruence are visible in Figure 1d, which shows at 0 nM tetramer a value of ~ 5 MFI while the same binding with 200 times more ligand is = 0 (which is meaningless in flow cytometry because 0 is when cells are not passed in the facs analyser). Moreover, while figure 1d at 0 nM tetramer shows a value of ~ 5 MFI, in figure 3a a similar experiment gives a background MFI of 2000!!

I feel also that the prolonged binding for tens of minutes even with 1 nM pMHC is due to background and that the specificity of the intracellular binding of the APA/1 is questionable.

5 - Extended data figure 1d. The profile presented is clearly a profile of protein optical density and not a test of tetramer binding. It is not possible that such test (in MFI) gives rise to a continuum curve, as it was monitored continuously. It is irritating that the authors make such a gross error, if not an intentional misrepresentation of data. Similarly, though clearly a misunderstanding of how to represent gel filtration profiles, it is physically impossible that dextran bleu used to provide the void volume of the gel filtration column, elutes at 0 ml. The void volume for this column (total volume of 6 ml as reported in material and methods) must be at ~ 2 ml

Reviewer #2 (Remarks to the Author):

The authors addressed my previous concerns in revised manuscript.
Few additional small comments:

The revised manuscript has few typos like on line 98 (should be 200-fold, not 40-fold) and line 62 needs reference or clarification. Also, I suggest to remove the "equilibrium" word from the new Figure 2 title "TCRs are in equilibrium between three states of conformations.", as "active" state is temporal and does not reach equilibrium as shown in Figure 2e.

Responses to Reviewers:

Reviewers' comments:

Reviewer #1 (Remarks to the Author):

There continue to be incoherence in the binding data and flow cytometry and lack of adequate controls.

1 - A main issue in figure 1 is the odd result that binding to cells of pMHC tetramers at 200 nM (Fig 1d) exposes the epitope of (APA/1), supposed to detect only the TCR intracellular tail, at a value (~ 150 MFI) lower than the the that attained by 1 nM (200 times lower ligand concentration!). Even if the authors are correct that there is a "lateral spreading" of TCR activation when just one is activated, using 200 times more ligand, one should see that many more supposed pre-existing clusters are activated. How to explain (for example at 2 min stimulation) that 1 nM tetramer gives a 10 MFI for binding with an MFI for APA/1 similar/reduced as compared to 200 nM tetramer giving an MFI for pMHC binding of 600 (60 times higher)?

This is one of the main points of the paper: very low concentrations of ligand (tetramer) saturate the response because it is not necessary to bind *all* TCRs but just to bind *all* TCR nanoclusters. Once all TCR nanoclusters are engaged, it does not make a difference if additional TCRs within the nanoclusters are recruited. In other words, 1 nM tetramer appears to saturate all TCR nanoclusters, not all TCRs. Indeed, lowering the concentration of tetramer to half (0.5 nM) results in lower APA1/1 MFI whereas increasing the concentration 2-fold (2 nM) results in the same APA1/1 MFI as with 1 nM (see Figure below).

The saturation of APA1/1 response at a concentration of 1 nM tetramer is in fact paralleled by the saturation of response, measured as CD69 expression, shown in Figure 1b at the concentration of 1 nM.

To illustrate the independence between total binding and the number of TCR nanoclusters in the Active state, we make use of the mathematical model described in the paper. Specifically, binding is a direct consequence of the presence of ligand and free TCRs but, on the contrary, the number of Active nanoclusters changes due to entropic changes in the nanocluster "landscape" induced by cross-linking of multivalent ligand. Thus, cross-linking is the main cause of the stabilization of the Active conformation of TCR nanoclusters and not the availability of free ligand.

2 - "Spreading" of conformational changes in a cluster should take milliseconds/seconds. But the data presented indicate that APA/1 epitope access reaches a maximum only at 6 minutes. There are suggestions of such mechanisms for membrane ion channels that indicate these lateral phenomena must be much faster. Besides, how can one explain that tyrosine phosphorylation of the engaged TCR for a given amount of ligand usually takes seconds to be detected and peaks at 1 minute, the appearance of the APA/1 epitope peaks only after 6 minutes?

As the referee says we also expect that fixation of the Active conformation for an *individual* TCR to be extremely fast. However, in the experiments such as Figure 1d we are not measuring the adoption of the Active conformation by *individual* TCRs, but the distribution of the Active conformation in the total population of TCRs in a per cell base. Binding of APA1/1 increases from minute 0 to minute 6 because there are more and more TCRs (we know that they are TCR nanoclusters) that are being engaged. So, between 0 and 6 minutes not *all* TCR nanoclusters have yet been engaged and that is why the Active conformation detected by APA1/1 in the entire TCR population continues to increase. This was indeed the expected scenario: more TCR engagement by tetramer results in a larger number of TCR clusters in the Active conformation. The unexpected result was the downhill behaviour after 6 minutes and this is another important result of the paper.

As discussed in the manuscript (and in the previous question), the presence of a maximum conformational change arises from a superposition of mechanisms, e.g., entropic changes in the state of the TCR nanocluster, binding, or cross-linking, in such a way that the timescale of the maximum cannot be traced to a single microscopic characteristic timescale. Actually, this is a generic property of dynamical systems even in very simple scenarios. For example, as shown in Currie et al., J. Royal Society: Interface 9, 2856 (2012), a monovalent ligand binding to receptor model with Ligand (L), Receptor (R) and Bound Complex (C), $R+L \leftrightarrow C$, has a characteristic timescale that is a complex combination of the kinetic parameters that describe the binding and the unbinding, but also depends on the dissociation constant, K_D , the total number of receptors (N_R) and the initial concentration of free ligand (q). In particular, if $z(t)$ describes the concentration of bound ligand (Complex), it can be shown that $z(t)$ satisfies the following equation:

$$z(t) = N_R \frac{\lambda_+ \lambda_- K_d}{\rho \eta} \frac{(1 - e^{-\Delta k_{off} t})}{(\lambda_+ - \lambda_- e^{-\Delta k_{off} t})}$$

where the timescale $\Delta \cdot k_{\text{off}}$ is not only proportional to the off-rate, k_{off} , but also to the parameter Δ ,

$$\eta = \frac{N_c N_R}{V N_A \rho} = \frac{M_R}{M_L}, \quad \lambda_{\pm} = \frac{1}{2} \left[1 + \frac{\rho}{K_d} (1 + \eta) \right. \\ \left. \pm \sqrt{\left[1 + \frac{\rho}{K_d} (1 - \eta) \right]^2 + \frac{4\rho}{K_d} \eta} \right] \quad \text{and} \quad \Delta = \lambda_+ - \lambda_-,$$

In our case, the mathematical model developed is not so simple, since it also includes the multivalent nature of the ligand and the conformational state of the TCR nanocluster. Thus, it is not so straightforward to derive conclusions about typical orders of magnitude of the characteristic timescales by looking at a single mechanism (as the one discussed by the referee).

3 - I had suggested to perform tetramer binding on control T cells. The authors did it but instead of using 1 nM pMHC tetramer they used 5 nM (figure 1f, with no numbers in the x-axes). Why so if they need to control for the data in figure 1d that depicts experiments carried at 1 nM? Because the numbers are at times so small (10 to 30) that one cannot see real differences. Besides, the binding data in figure 1f are meaningless without adequate statistics. There are ways to improve flow-cytometry data by labelling control cells with viable fluorescence dye and perform analyses at the same time on control samples.

We apologize for a mistake we made in the Text; in line 106 we wrote we used a tetramer concentration of 5 nM when instead we used 1 nM, since as the Reviewer says we aimed to control for a specific effect depending on the TCR that is expressed. In fact, although in the Results section we mentioned 5 nM, in the legends of Fig. 1f and Extended Data Fig. 1b we mentioned 1 nM.

On the other hand, we do not agree that small numbers are not significantly different. Why should one- or two-digit numbers not be significant? A difference between an MFI of 5 and an MFI of 20 can be significant if the repetition shows consistency. Furthermore, a difference between an MFI of 5 and an MFI of 20 can be perceived by a simple overlay plot. The Figure below illustrates this point. When OT1 cells are incubated with 1 nM tetramer, a clear CD8+tetramer+ population is detected; the CD8+ cells are however clearly negative if tetramer is not added (left and central panels). The difference between background and specific tetramer staining can be clearly seen when an overlay plot is used (right panel).

Finally, Figure 1f further illustrates the quantitative data shown in Extended Data Figure 1b, where there are statistically significant differences for times 0 and 5 for OT1 T cells, but not for HY T cells.

Examples of background variability and incongruence are visible in Figure 1d, which shows at 0 nM tetramer a value of ~ 5 MFI while the same binding with 200 times more ligand is = 0 (which is meaningless in flow cytometry because 0 is when cells are not passed in the facs analyser).

We believe the referee might have misinterpreted the different ranges of MFI values (scales) in the right y-axis in both conditions of Figure 1d. Whereas at 1 nM of tetramer the right y-axis represents a range between 0 and 60, at 200 nM the range is between 0 and 1200. Indeed, the MFI at time 0 in both conditions is the same because the 0 is common: MFI= 4.8 ± 0.3 (mean \pm sd) for the plot with 1 nM tetramer; 4.8 ± 0.3 for the plot with 200 nM.

We hope that the answer above satisfies the referee for that specific experiment. We also note that background fluorescence, in the absence of tetramer, is due to cell autofluorescence and that the MFI value of such background will depend on the voltage of the laser used in the specific experiment. If the voltage is low, the MFI could be 0 or even negative. So, if the MFI is 0, it does not necessarily mean absence of cells. The laser settings can change from one experiment to another, since this is a variable that we cannot control. For example, a laser loses potency with time of use. Another source of variability in the output of the MFI data results from the use of a FACS Diva versus a FACS Calibur. The first provides data in bi-exponential form and the latter in logarithmic form. Using a FACS Calibur, the autofluorescence background is set between 10^0 and 10^1 , in the FACS Diva the background is set at 10^2 . In addition, The FACS Calibur has a less potent laser than the FACS Diva and this might be responsible for the higher MFI values when samples were analyzed in the latter. Therefore, the “background variability” is perfectly normal. In all our experiments we included controls of unstained cells to set the background and controls of single-stained cells to compensate channels. Looking at the Materials and Methods section, we have now realized that we did not mention we used both types of flow cytometers. We have now mentioned this fact in the hope that it may help to explain inter-experimental differences in MFI values.

Moreover, while figure 1d at 0 nM tetramer shows a value of ~ 5 MFI, in figure 3a a similar experiment gives a background MFI of 2000!!

The referee might not have taken into account the extrapolation of the linear regression plot that at the intersection point with the y-axis gives a MFI value close to 2000 and the real data of MFI at time=0 which is close to 0 (red circle in the Figure below).

I feel also that the prolonged binding for tens of minutes even with 1 nM pMHC is due to background

The referee feels that the binding of 1 nM tetramer for minutes is background, yet the data robustly indicate that binding is clearly and statistically above background. The cause of background fluorescence can be categorized into three groups: (I) autofluorescence, (II) spectral overlap, and (III) undesirable antibody binding (Cytometry B Clin Cytom. 2009 Nov;76(6):355-64. doi: 10.1002/cyto.b.20485). Considerations for the control of background fluorescence in clinical flow cytometry. Hulspas R¹, O'Gorman MR, Wood BL, Gratama JW, Sutherland DR. In order to determine that the detected signals were above background we used the following different controls:

- (I) Unlabeled cells to set autofluorescence levels.
- (II) Single color controls for channel compensation and to correct for spectral overlap.
- (III) We have used cells from HY transgenic mice as negative controls to determine that tetramer binding to OT1 is specific to the TCR. This is shown in Extended Data Figure 1b.

The values plotted in the above Figure are the following:

OT1		0	1	2	5	8	15	30
time		131	221	247	279	320	444	964
		129	238	250	379	320	415	820
		136	238	226	276	324	466	782
t -test		7,67756E-05	0,000155	0,006134	1,74177E-07	3,17929E-05	0,0002	

HY		0	1	2	5	8	15	30
time		101	136	130	128	108	111	142
		123	118	129	125	114	106	117
		114	139	107	132	115	124	117
t -test		0,115721917	0,397211	0,962979	0,962978949	0,910340254	0,294099	

A two-tailed t-test shows significant differences between the set of values at t=0 and the sets of values at times 2, 5, 8, 15 and 30 minutes for OT1 T cells. However, none of the time points was significantly different to the time=0 for HY T cells. Furthermore, a two-way ANOVA test of the binding of OVAp tetramer to OT1 versus HY cells in the total time course shows a p-value <0.0001. Therefore, OVAp tetramer binding at 1 nM to OT1 cells is significantly above background at all time points.

In addition, we have analyzed by gel filtration (Extended Data Fig. 1d), as the referee suggested, the possible presence of tetramer aggregates as a source of continuous binding after tens of minutes and found none.

and that the specificity of the intracellular binding of the APA1/1 is questionable.

Same answer as above. All controls used show that APA1/1 binding is specifically induced by TCR triggering. This is what the other part of Extended Data Figure 1b shows:

There is a significant increase in MFI 5 minutes after stimulation over the t=0 in OT1 T cells but not in HY T cells, indicating that the increase in APA1/1 MFI is TCR-specific. In addition, we have data showing APA1/1 MFI after stimulation with 1 nM tetramer of WT OT1 cells and OT1 T cells bearing the C80G mutation in CD3epsilon (Figure 5b-d). Unlike WT cells, cells bearing the mutation in the ectodomain of CD3 epsilon, which prevents the outside-in transmission of conformational changes, do not respond to tetramer binding by increasing APA1/1 MFI (see Figure below). It would be hard to explain the difference between WT and C80G OT1 T cells if APA1/1 MFI was just a background effect.

In summary, we have done many experiments with APA1/1 and they show in a consistent manner a transient increase of APA1/1 MFI upon incubation with tetramer that is significantly above background. We are totally convinced this is not an artifact.

5 - Extended data figure 1d. The profile presented is clearly a profile of protein optical density and not a test of tetramer binding. It is not possible that such test (in MFI) gives rise to a continuum curve, as it was monitored continuously. It is irritating that the authors make such a gross error, if not an intentional misrepresentation of data. Similarly, though clearly a misunderstanding of how to represent gel filtration profiles, it is physically impossible that dextran bleu used to provide the void volume of the gel filtration column, elutes at 0 ml. The void volume for this column (total volume of 6 ml as reported in material and methods) must be at ~ 2 ml

First of all, we think that the referee's comment ***"It is irritating that the authors make such a gross error, if not an intentional misrepresentation of data"*** is not only out of place but disrespectful in a scientific review. The Reviewer might not have realized that we have used an Excel plot not showing the discrete data points (because there are many). In the Figure we used the following plot:

The alternative plot showing the data points is this one:

In summary, the data we plotted correspond to MFI values, not optical density. We did not, intentionally or unintentionally, misinterpret the experimental data.

In addition, we set the volume at which dextran blue elutes as volume 0 (another way of plotting the data). What is important is that in that fraction or in the fractions corresponding to a molecular weight >669 KDa, there is no binding activity, i.e., there are no aggregates that could alternatively explain different binding slopes as the referee suggested.

Reviewer #2 (Remarks to the Author):

The authors addressed my previous concerns in revised manuscript.
Few additional small comments:

The revised manuscript has few typos like on line 98 (should be 200-fold, not 40-fold) this has been corrected

and line 62 needs reference or clarification. we have included a reference to one of the many reviews on the subject

Also, I suggest to remove the "equilibrium" word from the new Figure 2 title "TCRs are in equilibrium between three states of conformations.", as "active" state is temporal and does not reach equilibrium as shown in Figure 2e.

The reviewer is absolutely right and we have now replaced "in equilibrium" by "transit": "TCRs are in transit between three states of conformations"

Reviewers' comments:

Reviewer #4 (Remarks to the Author):

The authors entertain an interesting idea explaining how engagement of few TCRs on the surface of T cells could lead to activation of many more TCR and T-cell activation. They proposed that “homotropic allosteric effects” constitute mechanism according which many TCR gets activated followed ligation of a few TCR.

Antibody specific for APA1/1 epitope on TCR epsilon chain was used to detect productive TCR engagement by cognate tetramers. Expression of activating markers CD69 and CD25 were used to monitor T-cell activation. Dynamic changes of these parameters induced by T cell stimulation by tetramers led the authors to propose 3- state model for TCR-mediated T cell response. The essence of the model is that productive engagement of a limited number of TCR with strong agonist pMHCs assembled into tetramers results in activation many more TCRs. Experimental data suggest that this process takes several minutes.

Although the data are interesting there are several concerns that needs to be addressed:

1. The authors shows that 1nM and 200 nM concentration of tetramer induces the same kinetics of APA1-1 antibody binding. However, 200-fold increase in the tetramer concentration should significantly accelerate the rate of the tetramer binding because kinetics of ligand-receptor interaction depends on ligand concentration. The time scale of the ligand binding to pre-existent TCR clusters is in a range of seconds at the conditions of the experiments. In addition, the tetramer binding should be temperature-dependent. Meanwhile, allosteric changes in protein structure that authors are talking about occurs in seconds. Thus, the observed kinetics of the changes in activation markers could be explained by reorganization of preexistent clusters including appearance of newly formed clusters after TCR crosslinking by multimeric ligand, which occurs in minutes. Clusters reorganization could include: (1) changes in the clusters size through recruitment of additional TCR or/and CD8 molecules, (2) changes of distance and orientation between TCR molecules within the clusters, (3) changes in the ratio of CD8 and TCR molecules inside of the clusters or/and changes in their relative positioning. These changes are mediated by cooperative binding of tetramer ligands to preexisting TCR-CD8 nanoclusters resulting in their reorganization, but not exclusively by allosteric changes in preexisted TCR lattice that occur in milliseconds. The classical definition of allosteric changes assumes spreading of conformational changes inside of receptor lattice (milliseconds range), while the authors experimental data suggest changes in super-positioning of TCR, which is very different process resulting in entropic reorganization of initial TCR nanoclusters structures. Thus, the presented data do not provide direct evidence for allosteric changes in TCR lattice, but also do

not exclude the possibility of allosteric changes in TCR lattice initiated by pMHC binding. Based on this consideration, I believe that the allosteric mechanisms could be discussed by authors, but should not be placed in the article title sending a very misleading message to the community.

Along this line signal spread model was proposed in earlier studies (BBA, 1853: 767-774, 2015; PNAS, 103: 16846-16851, 2006). The findings described in these papers should be reconciled and discussed by the authors.

2. The tetramer staining and the induction of T cell activation were performed in the presence of 0.02% sodium azide. Sodium azide is known as a toxic agent that interfered with cellular functional assays influencing particularly on cell respiration, endocytosis and receptors capping. It cannot be ruled out that presence of sodium azide in the buffer influenced on extent of time dependent APA-1 epitope exposure and tetramer staining especially at 37°C.

3. Nck interaction with Pro-rich moiety of CD3-epsilon cytoplasmic tail precedes TCR signaling. APA1-1 compete with Nck binding (Cell, 109: 901-912, 2002). Therefore, inhibition of APA-1 staining could reflect inhibition by Nck rather than induction of inhibitory state of TCR nanoclusters?

Along this line, tetramer staining in the center of the synapse could be diminished due to segregation of CD8 from TCR within the cSMAC zone.

4. Relatively long time of the transition to activation state may be due much lower stimulatory potency of pMHC tetramer whose pMHC arms are separated by much larger distances (PLoS ONE, 7: e41466, 2012) than those measured on the surface of live cells (Biophys J 74: 2184-2190, 1998; J Immunol 192: 52-58, 2014). Indeed, stimulatory potency of pMHC tetramers is very low and does not mimic potency of other model membrane clusters that mimic more closely physiological conditions (PLoS ONE, 7: e41466, 2012). Although the authors could see cooperative effect, but this may not be physiologically relevant.

5. Functional assays (CD69 and CD25 upregulation) were done with tetramer at 37°C for 24 hours. At these conditions, the peptide could partially dissociate into the assay media resulting in cross presentation to T cells. These may change the net result of the assay, so the appropriate controls are required.

6. The tetramer was eluted in fraction corresponding to molecular weight 150 to 669 kDa and several peaks of tetramer binding were observed in elution profiles. Keeping in mind that the tetramer MW is 350 kDa, the results show high heterogeneity of the sample. The heterogeneity of the tetramer preparations was shown previously (J Immunol., 187: 6281-6290, 2011) and revealed presence of pMHC monomers, dimers, trimers and tetramers. Low molecular weight

fluorophores Alexa647 would allow exploiting standard PAGE electrophoresis for analysis of tetramer preparations and would enhance brightness of T cell staining.

RESPONSE TO REVIEWER # 4's CONCERNS:

1. The authors shows that 1nM and 200 nM concentration of tetramer induces the same kinetics of APA1-1 antibody binding. However, 200-fold increase in the tetramer concentration should significantly accelerate the rate of the tetramer binding because kinetics of ligand-receptor interaction depends on ligand concentration.

We agree with the Reviewer with that idea although we had not emphasized it properly in the previous version. Indeed, Figure 1d shows that when OT1 T cells were incubated with 200 nM tetramer, maximal binding was reached in 6 minutes, whereas when incubated with 1 nM tetramer concentration, maximal binding was still not reached even after 20 min. We have taken the suggestion of the Reviewer and wrote a brief sentence in the Results section to mention the different kinetics of binding and its dependence on the concentration.

The time scale of the ligand binding to pre-existent TCR clusters is in a range of seconds at the conditions of the experiments. In addition, the tetramer binding should be temperature-dependent.

We agree with the Reviewer that binding should be temperature-dependent. Indeed, such effect is shown when comparing the MFI values for the tetramer when incubated at 0°C vs. 37°C at 1 nM concentration (Fig. 1d vs. Fig 1e).

Meanwhile, allosteric changes in protein structure that authors are talking about occurs in seconds. Thus, the observed kinetics of the changes in activation markers could be explained by reorganization of preexistent clusters including appearance of newly formed clusters after TCR crosslinking by multimeric ligand, which occurs in minutes.

The Reviewer is concerned about the possibility that incubation with tetramers does induce TCR clustering that could explain the enhanced binding of following tetramer molecules. To rule out that possibility we carried out most our experiments at 0°C. In these conditions we do not see any significant change in the size and distribution of TCR clusters either at the microscale (confocal microscopy) or the nanoscale (electron microscopy ; Extended Data Figure 6). By contrast, incubation with tetramer at 37°C does induce the formation of bigger TCR clusters that can be seen by confocal and electron microscopy (Extended Data Fig. 6). Interestingly, in spite the increased clustering detected at 37°C, the optimum of the Active conformation, detected with APA1/1, does not change (Fig6d vs. Fig. 6e). All these data support the idea that the detected optima and binding cooperativity (especially at 0°C) are not related to increased size of the pre-existing TCR nanoclusters.

Clusters reorganization could include: (1) changes in the clusters size through recruitment of additional TCR or/and CD8 molecules,

As discussed in the previous point, we show that there is not such a change in the size of TCR clusters at 0°C (Extended Data Fig. 6).

In addition, in Extended Data Fig. 4 we show that blocking CD8-MHC interaction with an anti-CD8 or using a mutant tetramer does not affect the time at which tetramer binding is enhanced. We think the data rule out an effect for CD8 as responsible for the cooperativity effect within TCR nanoclusters.

(2) changes of distance and orientation between TCR molecules within the clusters,

We agree with this interpretation since it points out to the adoption of the Active conformation of the TCR, manifested as re-orientation of TCRs within a cluster. Proof that the Active conformation is required is that the C80G mutant, which abrogates adoption of the Active conformation prevents cooperativity effects on ligand binding (Fig. 5d).

(3) changes in the ratio of CD8 and TCR molecules inside of the clusters or/and changes in their relative positioning. These changes are mediated by cooperative binding of tetramer ligands to preexisting TCR-CD8 nanoclusters resulting in their reorganization, but not exclusively by allosteric changes in preexisted TCR lattice that occur in milliseconds.

Again, the lack of effect of CD8-MHC binding blockade (Extended Data Fig. 4) argues against a participation of CD8 in the phenomenon of TCR cooperativity on ligand binding that we are reporting.

In addition, we show through disruption of TCR nanoclusters with MbCD that preexisting TCR lattices are required for the cooperativity effect that we detect using tetramers labeled with different fluorophores. Please, note that we detect this effects in minutes since the techniques used do not allow to probe shorter timescales, i.e. to see some detectable binding we need to incubate for some time in order to detect sufficient binding in the entire population of cells. We believe the changes we detect in minutes in the entire population of TCRs and cells respond to the accumulation of events happening in milliseconds but since we cannot measure single TCR nanoclusters and single cells, we cannot shorten our time scale to that level of resolution. So, TCR allostery is rather an explanation inferred from the cooperative effects on ligand binding that we detect at 0°C and not the result of a direct measurement.

The classical definition of allosteric changes assumes spreading of conformational changes inside of receptor lattice (milliseconds range), while the authors experimental data suggest changes in super-positioning of TCR, which is very different process resulting in entropic reorganization of initial TCR nanoclusters structures. Thus, the presented data do not provide direct evidence for allosteric changes in TCR lattice, but also do not exclude the possibility of allosteric changes in TCR lattice initiated by pMHC binding. Based on this consideration, I believe that the allosteric mechanisms could be discussed by authors, but should not be placed in the article title sending a very misleading message to the community.

Entropic reorganization of initial TCR nanoclusters is an expression of conformational change spreading. However, since the Reviewer is right with our inability to show changes that take place in the timescale of milliseconds, we are happy to change the Title of the manuscript and propose the more balanced title: "A window of opportunity for **cooperativity** in the T cell receptor"

Along this line signal spread model was proposed in earlier studies (BBA, 1853: 767-774, 2015; PNAS, 103: 16846-16851, 2006). The findings described in these papers should be reconciled and discussed by the authors.

We have referenced both articles and have now discussed the QD article in PNAS about cooperativity between non-cognate and cognate pMHC complexes on CD8 T cell activation. Although the PNAS paper suggests help of non-cognate in T cell response to cognate pMHC, something reported one year earlier (Nature 434: 238-243, 2005; Ref. 36), looking at the immunological synapse, the PNAS paper examines the effect on global T cell activation and does not tell about how a few binding events favor subsequent ones, as we do in the present paper. In fact, the QD data in the PNAS paper could very well be explained by just an avidity effect.

2. The tetramer staining and the induction of T cell activation were performed in the presence of 0.02% sodium azide. Sodium azide is known as a toxic agent that interfered with cellular functional assays influencing particularly on cell respiration, endocytosis and receptors capping. It cannot be ruled out that presence of sodium azide in the buffer influenced on extent of time dependent APA-1 epitope exposure and tetramer staining especially at 37°C.

Our apologies for the potential confusion regarding this matter. Incubations at 0°C were carried out in the presence of sodium azide to further inhibit any possible metabolism but incubations at 37°C were carried out in buffered RPMI medium without azide. We have now remarked this difference in the Methods section of the manuscript.

3. Nck interaction with Pro-rich moiety of CD3-epsilon cytoplasmic tail precedes TCR signaling. APA1-1 compete with Nck binding (Cell, 109: 901-912, 2002). Therefore, inhibition of APA-1 staining could reflect inhibition by Nck rather than induction of inhibitory state of TCR nanoclusters?

We carried out most experiments at 0°C to inhibit the 2D and 3D movement of membrane and cytoplasmic proteins. Again, the insensitivity of APA1/1 exposure to temperature (Fig.1d vs. Fig. 1e) argues against a competition by intracellular Nck since this should be expected to be enhanced at 37°C compared to 0°C. In addition, the Cell paper showed that APA1/1 outcompeted Nck binding. Finally, in another paper (J Immunol, 192: 2042-2053, 2014) we calculated the affinity of Nck for the CD3-epsilon tail as 0.5 μM, far from the typical affinities of antibodies.

Along this line, tetramer staining in the center of the synapse could be diminished due to segregation of CD8 from TCR within the cSMAC zone.

That possibility is something we tried to rule out by using a tetramerized biotin-labeled anti-CD3 antibody (Extended Data Fig. 8). A tetrameric anti-CD3 antibody is much bigger than a MHC tetramer and we do not see such exclusion from the cSMAC zone. So, we do not think it is a problem of not accessibility to the cSMAC.

4. Relatively long time of the transition to activation state may be due much lower stimulatory potency of pMHC tetramer whose pMHC arms are separated by much larger distances (PLoS ONE, 7: e41466, 2012) than those measured on the surface of live cells (Biophys J 74: 2184-2190, 1998; J Immunol 192: 52-58, 2014). Indeed, stimulatory potency of pMHC tetramers is very low and does not mimic potency of other model membrane clusters that mimic more closely physiological conditions (PLoS ONE, 7: e41466, 2012). Although the authors could see cooperative effect, but this may not be physiologically relevant.

We have used a pMHC oligomer of completely different geometry, a dextramer (Extended Data Fig. 5) to rule out the possibility that any of effects are due to the peculiar geometry of tetramers. Nonetheless, the most important piece of information arguing in favor of a physiological process is that incubation with antigen-loaded dendritic cells also causes a transient enhancement of pMHC binding (Fig. 7a). We show that in response to antigen presentation by professional antigen-presenting cells, in the absence or in the presence of the Src tyrosine kinase inhibitor PP2, there is a transient upregulation of tetramer binding. In this case, the stimulus is physiological and only the reading system (binding of tetramer) is artificial.

5. Functional assays (CD69 and CD25 upregulation) were done with tetramer at 37°C for 24 hours. At these conditions, the peptide could partially dissociate into the assay media resulting in cross presentation to T cells. These may change the net result of the assay, so the appropriate controls are required.

The Reviewer is right, after such a long incubation (Fig. 1b) binding data of pMHC tetramer are not reliable. That is why we did the following experiments at 0°C for a much shorter time (1 hour, Fig. 1c). Both results, indicate that the concentrations of tetramer used later (1-5 nM), and which are optimal for CD69 and CD25 upregulation, are very far from saturating the TCR.

6. The tetramer was eluted in fraction corresponding to molecular weight 150 to 669 kDa and several peaks of tetramer binding were observed in elution profiles. Keeping in mind that the tetramer MW is 350 kDa, the results show high heterogeneity of the sample. The heterogeneity of the tetramer preparations was shown previously (J Immunol., 187: 6281-6290, 2011) and revealed presence of pMHC monomers, dimers, trimers and tetramers. Low molecular weight fluorophores Alexa647 would allow exploiting standard PAGE electrophoresis for analysis of tetramer preparations and would enhance brightness of T cell staining.

The expected size of the pMHC-APC tetramer is 242 kDa(tetramer)+270 kDa (APC), in total around 520 kDa, which is not far from the upper size measured by gel filtration (Extended Data Fig. 1). The gel filtration assay was proposed by Reviewer #1 to show if there were larger aggregates of tetramers (valency >4) that could explain changes in binding with time. The results of Extended Data Fig. 1 showed that this was not the case. The gel filtration assay shows that there are tetramers and oligomers of lower valency, as Reviewer #4 is indicating, but not higher valency ones, as Reviewer #1 had previously suggested. We have now referred to the paper suggested by the Reviewer and included an additional paper in which a gel filtration assay shows a size for the MHC tetramer in the range of ours.

Reviewers' comments:

Reviewer #4 (Remarks to the Author):

There are still some issues to be address:

1. The authors agree that the higher concentration of the tetramer accelerate the tetramer binding. It should increase the rate of TCR crosslinking and consequently the rate of APA1 epitope expose. However, neither changing concentration of the tetramer nor the temperature increase from 0oC to 37oC influenced on time required for maximal APA1 antibody site exposure. Perhaps, the changes are there but they are too small to detect by utilized method.
2. The proposed by authors model suggests that conformational changes do not depends on temperature and “assume that it is due to purely entropic effect”. Conformational changes in protein structure do depend on temperature as well as entropy contribution into energetic landscape, and the question is only about extent. The authors should not be casual in making statements.
3. Molecular weight APC is 105 kDa, but not 270 kDa as authors stated in the manuscript. Does this mean that the authors use Streptavidin-APC conjugate labeled with 2.5 APC molecules per streptavidin in average? If so, it is hard to believe that the streptavidin has four and even three available biotin binding sites. Thus, most of pMHC-Streptavidin conjugates are likely to be dimeric.
4. Line 592 and 606: errors are in equation numeration
5. Supplementary Fig. 6: p value should be indicated. Are there any clusters have been observed for sizes 10-15, 16-20 and 21-25 for the cells untreated with tetramer? If so, the tetramer binding induces TCR clustering not only at 37oC but also at 0oC. At 0oC the membrane is not “frozen”, and membrane proteins could still diffuse.
6. Supplementary Fig. 8. Position of target cell should be properly indicated.

Responses to Reviewers:

Reviewers' comments:

Reviewer #4 (Remarks to the Author):

We thank Reviewer #4 for his/her thoughtful comments on the manuscript and for the help to improve it.

There are still some issues to be address:

1. The authors agree that the higher concentration of the tetramer accelerate the tetramer binding. It should increase the rate of TCR crosslinking and consequently the rate of APA1 epitope expose. However, neither changing concentration of the tetramer nor the temperature increase from 0oC to 37oC influenced on time required for maximal APA1 antibody site exposure. Perhaps, the changes are there but they are too small to detect by utilized method.

We agree with the Reviewer that accelerated tetramer binding should result in faster APA1/1 epitope exposure. However as the Reviewer proposes the methods we use to detect changes in kinetics are not sufficiently precise. This handicap probably derives from the use of populations of T cells and not single T cells. Binding data refer to averages in the cell population and since binding to cells is not synchronized MFI values fluctuate. Since tetramer binding is followed by a fixation, a permeabilization and several incubation steps with antibodies we cannot follow binding and the Active conformation in real time. Nonetheless, optimal times for the entire population of 4-8 minutes are repeatedly generated and are significantly above time 0 and longer time points

2. The proposed by authors model suggests that conformational changes do not depends on temperature and “assume that it is due to purely entropic effect”. Conformational changes in protein structure do depend on temperature as well as entropy contribution into energetic landscape, and the question is only about extent. The authors should not be casual in making statements.

From the modeling perspective, our assumption is that, taking into account the fact that at 0°C and 37°C there are not significant differences in the effect, the temperature should be a mild factor in the free energy. We totally agree with the Reviewer that conformational changes are, in general, dependent on temperature. We have clarified this point in the new version to emphasize the “almost independence” rather than “purely entropic”. We thank the Reviewer for this remark.

3. Molecular weight APC is 105 kDa, but not 270 kDa as authors stated in the manuscript. Does this mean that the authors use Streptavidin-APC conjugate labeled with 2.5 APC molecules per streptavidin in average? If so, it is hard to believe that the streptavidin has four and even three available biotin binding sites. Thus, most of pMHC-Streptavidin conjugates are likely to be dimeric.

The Reviewer is right and most citations to the molecular weight of allophycocyanin refer to it as a 105 kDa protein. We have changed the main Text of the manuscript accordingly. However, we used tetramers from TCmetrix that was commercializing “real” tetramers elaborated with “tetra-grade” streptavidin, such as Immanuel Luescher and Phillippe Guillaume were referring to in the enclosed paper (pdf for Reviewer’s discretion). In any case, the use of a completely different multimer (dextramer, Suppl. Fig. 5) reproduced the time optimum detected with the tetramers.

4. Line 592 and 606: errors are in equation numeration

We have corrected the number sequence.

5. Supplementary Fig. 6: p value should be indicated. Are there any clusters have been observed for sizes 10-15, 16-20 and 21-25 for the cells untreated with tetramer? If so, the tetramer binding induces TCR clustering not only at 37oC but also at 0oC. At 0oC the membrane is not “frozen”, and membrane proteins could still diffuse.

Yes, there were also clusters in cells not incubated with tetramer of 10-15, 16-20 and 21-25 gold particles but the percentage was quite low although the differences with the distribution with cells incubated with tetramer at 0°C were not significant. We have introduced a new Figure in which we have increased the number of cells measured per condition. We have represented the p value of the significant differences (for clusters 26-28 and clusters >30); all other clusters were not significantly different.

6. Supplementary Fig. 8. Position of target cell should be properly indicated.

We have now marked the position of the antigen presenting cell in the Figure.